# Learning in Prophet Inequalities with Noisy Observations

**Jung-hun Kim**[1,3]      **Vianney Perchet**[1,2,3]
[1]CREST, ENSAE, IP Paris
[2]Criteo AI Lab
[3]FairPlay joint team
junghun.kim@ensae.fr     vianney.perchet@normalesup.org

## Abstract

We study the prophet inequality, a fundamental problem in online decision-making and optimal stopping, in a practical setting where rewards are observed only through noisy realizations and reward distributions are unknown. At each stage, the decision-maker receives a noisy reward whose true value follows a linear model with an unknown latent parameter, and observes a feature vector drawn from a distribution. To address this challenge, we propose algorithms that integrate learning and decision-making via lower-confidence-bound (LCB) thresholding. In the i.i.d. setting, we establish that both an Explore-then-Decide strategy and an $\varepsilon$-Greedy variant achieve the sharp competitive ratio of $1 - 1/e$, under a mild condition on the optimal value. For non-identical distributions, we show that a competitive ratio of $1/2$ can be guaranteed against a relaxed benchmark. Moreover, with limited window access to past rewards, the tight ratio of $1/2$ against the optimal benchmark is achieved.

## 1 Introduction

The prophet inequality is a fundamental problem in online decision-making and optimal stopping (Hill & Kertz, 1992). A decision-maker (or gambler) sequentially observes a stream of random variables (or rewards) revealed one by one and must decide at each stage whether to *accept the current value and stop*, or *continue to the next stage*. The benchmark is the *prophet*, an omniscient agent who knows all realizations in advance. The objective of the gambler is to design an online stopping rule whose expected payoff is competitive with that of the prophet, aiming to maximize the competitive ratio. This framework has been extensively studied, owing to its rich mathematical structure and broad applications such as posted-price mechanisms (Lucier, 2017), online ad allocation (Alaei et al., 2012), and hiring processes in labor markets (Arsenis & Kleinberg, 2022).

Classical work has established sharp guarantees when the underlying distributions are known. In particular, Samuel-Cahn (1984) showed that a single-threshold strategy achieves the optimal competitive ratio $1/2$ for independent but non-identical distributions. In the i.i.d. case, Hill & Kertz (1982) established the ratio $1 - 1/e$, which is the optimal under single threshold/quantile rule (Correa et al., 2019b; 2017).

Crucially, all of these guarantees assume full knowledge of the underlying reward distributions—an assumption that is rarely met in practice. This has motivated a recent line of work on prophet inequalities with *unknown* distributions (Correa et al., 2019a; 2020; Goldenshluger & Zeevi, 2022; Immorlica et al., 2023). A striking negative result of Correa et al. (2019a) is that, in this setting, *there is no meaningful online learning of the distribution*: the best achievable competitive ratio is $1/e \approx 0.368$, attained by the classical secretary algorithm. Achieving a larger ratio of $1 - 1/e \approx 0.632$ for i.i.d. setting requires $\Theta(n)$ additional *offline* samples (Correa et al., 2019a). Similarly, in the non-i.i.d. case with unknown distributions, obtaining a $1/2$ competitive ratio also necessitates $\Theta(n)$ offline samples (Rubinstein et al., 2019). More broadly, the literature indicates that nontrivial learning is impossible without either $\Theta(n)$ offline data or access to multiple independent repetitions of the decision problem, as in regret-minimization formulations rather than single-sequence competitive-ratio guarantees (Gatmiry et al., 2024; Liu et al., 2025). These requirements substantially constrain appli-

cability in real-world settings, where large offline datasets and repeated independent sequences are often unavailable.

In this work, we study the prophet inequality in a novel and practical setting, in which reward distributions are *unknown* without available offline reward samples, and furthermore, at each stage only a *noisy* realization of the random variable is observed. Instead, the decision-maker has access to observable feature vectors drawn from distributions, and the rewards follow a linear model with an unknown latent parameter. This structural information enables estimation of the reward distribution and fundamentally distinguishes our setting from the classical unknown-distribution model (Correa et al., 2019a; Rubinstein et al., 2019). This feature-based formulation is motivated by applications such as online advertising, hiring, and recommendation systems, where contextual information (e.g., ad profiles, candidate attributes, or item descriptions) and noisy feedback are observable, while the underlying reward distributions remain unknown.

**Exploration under stopping.**   In our setting, exploration is tied to the stopping decision: once the decision-maker chooses *stop and accept*, the process ends and no further information can be gathered. As a result, continuing to explore serves two intertwined purposes. First, it provides additional noisy, feature-dependent observations that help *learn* the unknown reward model (and thereby reduce statistical uncertainty). Second, it keeps the option open to *encounter new opportunities*, i.e., potentially higher future rewards. Since stopping irrevocably terminates both learning and opportunity, the decision-maker faces a delicate tradeoff between gathering more information and waiting for better candidates versus committing early to exploit a seemingly promising one under uncertainty.

We study learning-based stopping policies that couple estimation and stopping under noisy observations and feature feedback, and provides performance guarantees for our proposed policy. The main contributions are as follows:

**Summary of Contributions.**

- **Noisy linear prophet model with online learning.** Motivated by practical scenarios, we introduce a prophet-inequality setting in which the gambler observes only noisy rewards together with contextual features, while the latent rewards follow an *unknown* linear model. This structure enables an online-learning approach to prophet inequalities.

- **i.i.d. guarantees.** In the i.i.d. case, we propose learning-based stopping policies based on lower-confidence-bound (LCB) thresholding and prove that they achieve the sharp competitive ratio $1 - 1/e$ against the optimal prophet benchmark under a mild condition on the optimal value.

- **Non-identical guarantees.** For non-identical distributions, we analyze an algorithm that attains a $1/2$ competitive ratio against a relaxed benchmark. Moreover, with limited window access to past rewards, our method achieves the optimal competitive ratio $1/2$ against the standard prophet benchmark.

## 2   RELATED WORK

**Prophet Inequalities under Known Reward Distributions.**   The study of prophet inequalities originates from Krengel & Sucheston (1977; 1978). A key milestone was established by Samuel-Cahn (1984), who showed that a single-threshold strategy achieves the optimal competitive ratio of $1/2$ in the case of independent but non-identical distributions. In the order-selection variant, where the gambler can choose the order of arrivals, Chawla et al. (2010) achieved a ratio of $1 - 1/e$. For the i.i.d. case, Hill & Kertz (1982) established a ratio of $1 - 1/e$, which is optimal under single (time-invariant) threshold/quantile policies (Correa et al., 2019b; 2017). Subsequent work showed that this bound can be surpassed by allowing adaptive strategies (Abolhassani et al., 2017; Correa et al., 2017).

Extending beyond exact observations, Assaf et al. (1998) demonstrated that analogous guarantees remain valid under noisy observations, though only with respect to a Bayesian version of the prophet benchmark, which is weaker than the classical one. Indeed, under noisy observations, any nontrivial guarantee with respect to the classical benchmark becomes impossible without additional

structural assumptions, as we will show later. Finally, all of these results assume full knowledge of the underlying reward distributions—an assumption rarely satisfied in practical applications.

**Prophet Inequalities under Unknown Reward Distributions.**  To address the limitation of knowing distributions, recent work has studied prophet inequalities under unknown reward distributions (Correa et al., 2019a; 2020; Goldenshluger & Zeevi, 2022; Immorlica et al., 2023; Gatmiry et al., 2024; Li et al., 2022). For the i.i.d. setting, Correa et al. (2019a) showed that a competitive ratio of $1/e(\approx 0.368)$ can be achieved by the classical optimal algorithm for the secretary problem as the horizon grows. To obtain the higher ratio of $1 - 1/e(\approx 0.632)$, however, $\Theta(n)$ additional offline reward samples are required. Building on this, Goldenshluger & Zeevi (2022) showed that an asymptotic ratio approaching 1 is attainable, but only for fixed distributions whose maxima lie in the Gumbel or reverse-Weibull domains of attraction as the horizon grows.

The case of unknown non-identical distributions has been examined in Kaplan et al. (2020); Rubinstein et al. (2019); Gatmiry et al. (2024); Liu et al. (2025). However, achieving a $1/2$ competitive ratio requires $\Theta(n)$ offline samples. Although Gatmiry et al. (2024) and Liu et al. (2025) avoid offline samples, their setting involves repeated sequences of rounds rather than a single sequence. This repetition allows information to be aggregated across rounds, making the learning problem tractable under bandit feedback. In contrast, our setting involves only a single sequence, and is therefore fundamentally different. Prophet inequalities under unknown and non-independent distributions were also studied in Immorlica et al. (2023), achieving a ratio of $1/(2er)$ for $r$-sparse correlated structures, but their model still assumes distributional knowledge of the independent components of the rewards.

In contrast, we study a novel and practical setting that targets the optimal prophet under noisy reward observations and unknown reward distributions without available offline reward samples. Instead, we exploit observable feature vectors and their distribution, a setting motivated by real-world applications where feature information is available but the reward distribution is unknown.

## 3 PROBLEM STATEMENT

We consider $n$ non-negative random variables (or rewards) $X_1, \ldots, X_n$, where each $X_i$ is independently drawn from an *unknown* distribution $\mathcal{D}_i$. In particular, we assume that

$$X_i = x_i^\top \theta, \quad i \in [n],$$

where $x_i \in \mathbb{R}^d$ is a feature vector drawn independently from a distribution $\mathcal{D}_{x,i}$ to which the learner has sampling access, and $\theta \in \mathbb{R}^d$ is an *unknown* latent parameter. Since $\theta$ is unknown, the induced distributions $\mathcal{D}_i$ of the $X_i$ are also unknown to the gambler.

At each stage $i$, the gambler does not observe $X_i$ directly. Instead, it observes a *noisy* measurement

$$y_i = X_i + \eta_i,$$

where the noise $\eta_i$ is i.i.d drawn from a $\sigma$-sub-Gaussian distribution for $\sigma > 0$. The noisy observations $y_1, y_2, \ldots$ are revealed sequentially.

After observing $(y_i, x_i)$ at stage $i$, the gambler must make an irrevocable decision to either accept the current index and stop, or continue to the next stage. We denote by $\tau \in [n+1]$ the stopping time at which an index is accepted, with $\tau = n + 1$ corresponding to the event that the gambler rejects all observations, in which case the realized payoff is set to $X_{n+1} = 0$.

The gambler's expected payoff is $\mathbb{E}[X_\tau]$. As a benchmark, we consider the prophet—an omniscient decision maker who knows all values $X_1, \ldots, X_n$ in advance—which achieves $\mathbb{E}\left[\max_{i \in [n]} X_i\right]$. The goal of the stopping policy $\tau$ is to maximize the *asymptotic competitive ratio* against the prophet, given by

$$\liminf_{n \to \infty} \frac{\mathbb{E}[X_\tau]}{\mathbb{E}[\max_{i \in [n]} X_i]}.$$

**Notation.**  For a square matrix $M$, $\lambda_{\min}(M)$ denotes its minimum eigenvalue.

We consider regularization conditions as follows.

**Assumption 3.1.** *There exists $S > 0$ such that $\|\theta\|_2^2 \le S$.*

**Assumption 3.2.** *There exists $L > 0$ such that, for all $i \in [n]$ and $x \sim \mathcal{D}_{x,i}$, $\|x_i\|_2^2 \le L$*

Our regularization assumptions are standard in the online linear learning literature (Abbasi-Yadkori et al., 2011; Ruan et al., 2021; Liu et al., 2025).

## 4 THE I.I.D. SETTING

Here, we focus on the case where all reward distributions are identical, i.e., $\mathcal{D}_i = \mathcal{D}$ for every $i \in [n]$. This holds, for instance, when the feature distributions are identical across stages, i.e., $\mathcal{D}_{x,i} = \mathcal{D}_x$ for $i \in [n]$. Under this setting, we propose algorithms and analyze their competitive ratios.

### 4.1 EXPLORE-THEN-DECIDE WITH LCB THRESHOLDING

We first propose an algorithm (Algorithm 1) based on Explore-then-Decide with lower-confidence-bound (LCB) thresholding. To address the unknown distribution $\mathcal{D}$, the algorithm begins with a forced exploration phase of length $\ell_n$, provided as an input. After the exploration phase, the algorithm enters the decision phase, where it computes an LCB for the reward at each stage and applies an LCB-based thresholding rule to decide whether to stop or continue. This thresholding mechanism explicitly accounts for estimation uncertainty. The full procedure is detailed below.

#### 4.1.1 STRATEGY

**Exploration.** With setting $\ell_n = o(n)$, during the first $\ell_n$ stages, we collect pairs of noisy rewards $y_t$ and features $x_t$ at each stage $t$. Using these observations, we estimate the unknown parameter $\theta$ as $\hat{\theta} = V^{-1} \sum_{t=1}^{\ell_n} y_t x_t$, where $V = \sum_{t=1}^{\ell_n} x_t x_t^\top + \beta I_d$ for a constant $\beta > 0$.

After this exploration phase, the algorithm enters the decision phase, where it determines at each stage whether to stop or continue. The details regarding LCB Thresholding are given below.

**Lower Confidence Bound (LCB).** We define the lower confidence bound for $X_i$ as

$$X_i^{\mathrm{LCB}} = x_i^\top \hat{\theta} - \xi(x_i), \tag{1}$$

where $\xi(x_i) := \sqrt{x_i^\top V^{-1} x_i}(\sigma\sqrt{d \log(n + n\ell_n L/d\beta)} + \sqrt{S\beta})$.

**Decision with LCB Threshold.** Using the CDF of $\mathbb{P}_{z \sim \mathcal{D}_x}(Z^{\mathrm{LCB}} \le \alpha|\hat{\theta}, V)$ where $Z^{\mathrm{LCB}} = z^\top \hat{\theta} - \xi(z)$, we set threshold $\alpha$ s.t.

$$\mathbb{P}_{z \sim \mathcal{D}_x}(Z^{\mathrm{LCB}} \le \alpha|\hat{\theta}, V) = 1 - \frac{1}{n} \tag{2}$$

The algorithm stops at stage $i > \ell_n$ if $X_i^{\mathrm{LCB}} \ge \alpha$, in which case we set $\tau = i$. By definition, if no stopping occurs throughout the horizon, we set $\tau = n + 1$. Although Algorithm 1 is presented assuming oracle access to $D_x$ for clarity, the quantile defining the threshold $\alpha$ can be approximated arbitrarily well via Monte Carlo sampling from $D_x$.

#### 4.1.2 THEORETICAL ANALYSIS

In this setting, a fundamental limitation emerges due to noisy observations. In fact, it is possible to construct instances where the observation noise drives the competitive ratio to a trivial limit, as formalized below (see Appendix A.1 for the proof).

**Proposition 4.1.** *There exists a bounded i.i.d. distribution for $(X_i)_{i=1}^n$ together with an observation noise model such that, for any stopping policy $\tau$ based on the observations,*

$$\lim_{n \to \infty} \frac{\mathbb{E}[X_\tau]}{\mathbb{E}[\max_{i \in [n]} X_i]} = 0.$$

---

**Algorithm 1** Explore-Then-Decide with LCB Thresholding (ETD-LCBT)

**Input:** Exploration length $\ell_n$; regularization parameter $\beta$
**Output:** Stopping time $\tau$

1 **for** $i = 1, \ldots, n$ **do**
2      **if** $i \le \ell_n$ **then**
3          Observe $(y_i, x_i)$
4          **if** $i = \ell_n$ **then**
5              $V \leftarrow \sum_{t=1}^{\ell_n} x_t x_t^\top + \beta I_d$; $\hat{\theta} \leftarrow V^{-1} \sum_{t=1}^{\ell_n} y_t x_t$
6              Compute $\alpha$ from (2) (or (5) for non-i.i.d.)
7      **else**
8          Observe $(y_i, x_i)$
9          Compute $X_i^{\mathrm{LCB}}$ from (1)
10          **if** $X_i^{\mathrm{LCB}} \ge \alpha$ **then**
11              Stop and set $\tau \leftarrow i$

---

The trivial outcome in Proposition 4.1 explains why Assaf et al. (1998) studied a Bayesian version of the prophet inequality rather than the classical one ($\mathbb{E}[\max_{i \in [n]} X_i]$) under the noisy observation. As Proposition 4.1 shows, even with full knowledge of the reward distribution, no algorithm can avoid this collapse to a trivial competitive ratio. To overcome this fundamental challenge—both in targeting the classical prophet under noisy observation and in the presence of an unknown latent parameter in the reward distribution—we later impose a mild non-degeneracy condition on reward scaling.

For notational convenience, let $\lambda = \lambda_{\min}\left(\mathbb{E}_{x \sim \mathcal{D}_x}[xx^\top]\right)$, the minimum eigenvalue of the covariance matrix of the feature distribution. Without loss of generality, we restrict attention to the case $\lambda > 0$, ensuring non-degeneracy of the feature covariance. Under this notation, we can now state our main guarantee on the competitive ratio (see Appendix A.2 for the proof).

**Theorem 4.2.** *The stopping policy $\tau$ of Algorithm 1 with $\ell_n = o(n)$, $\ell_n = \omega(\frac{L \log d}{\lambda})$, and a constant $\beta > 0$, achieves an asymptotic competitive ratio of*

$$\liminf_{n \to \infty} \frac{\mathbb{E}[X_\tau]}{\mathbb{E}[\max_{i \in [n]} X_i]} \ge 1 - \frac{1}{e} - \limsup_{n \to \infty} \varepsilon_n,$$

*where*

$$\varepsilon_n := \mathcal{O}\left( \frac{1}{\mathbb{E}\left[\max_{i \in [n]} X_i\right]} \sqrt{\frac{L(\sigma^2 d + S) \log(Ln)}{\lambda \ell_n}} \right).$$

This result highlights the critical role of the optimal value $\mathrm{OPT} := \mathbb{E}[\max_{i \in [n]} X_i]$ in determining the competitive ratio under noisy learning. As shown in Proposition 4.1, without further structural assumptions, the competitive ratio can collapse to zero. To circumvent this issue, we impose a non-degeneracy condition on reward scaling, specifically on the growth of OPT, which ensures learnability under noise and allows us to recover the sharp bound established in Theorem 4.2.

**Corollary 4.3.** *We set $\ell_n = \frac{L(\sigma^2 d + S)}{\lambda} f(n) \log(Ln)$ for some function $f(n)$ (e.g., $f(n) = \Theta(\log^p n)$ for $p > 0$, or $\Theta(n^q)$ for $0 < q < 1$) satisfying $\ell_n = o(n)$. If $\mathrm{OPT} = \omega(1/\sqrt{f(n)})$, then the stopping policy $\tau$ of Algorithm 1 achieves an asymptotic competitive ratio of*

$$\liminf_{n \to \infty} \frac{\mathbb{E}[X_\tau]}{\mathbb{E}[\max_{i \in [n]} X_i]} \ge 1 - \frac{1}{e}.$$

The non-degeneracy condition of $\mathrm{OPT} = \omega(1/\sqrt{f(n)})$ in Corollary 4.3 is mild in practice. For example, by using $f(n) = n^{2/3}$ for setting $\ell_n$, the requirement is satisfied in most applications since OPT typically remains bounded away from zero. In particular, it suffices that $\mathrm{OPT} \ge C$ for some constant $C > 0$ and all sufficiently large $n$.

Our competitive ratio of $1 - 1/e$ matches that of Hill & Kertz (1982) in the known i.i.d. setting and that of Correa et al. (2019a) in the unknown i.i.d. setting but with $\Theta(n)$ additional offline reward

samples. Without such samples, only a $1/e$ ratio can be guaranteed (Correa et al., 2019a), which is strictly weaker than our result. Moreover, because rewards in our setting are observed only through noisy realizations, these prior guarantees no longer apply. Also note that $1 - 1/e$ is the best possible competitive ratio attainable by single quantile-threshold policies (Correa et al., 2019b; 2017), and our algorithm provides a close approximation to this class of policies.

Importantly, while Correa et al. (2019a) show that $1/e$ is optimal for unknown distributions without sufficiently many offline reward samples of $\Omega(n)$, we demonstrate that by exploiting feature information under structural assumptions, the sharp bound of $1 - 1/e$ can in fact be achieved. Moreover, our analysis accommodates distributions whose support grows with the horizon $n$ (e.g., $L = \sqrt{n}$ when setting $f(n) = \log n$ in Corollary 4.3), so that both the support and the variance of $\mathcal{D}$ may diverge as $n \to \infty$. This highlights that our framework is not restricted to the fixed distributional domains considered in Goldenshluger & Zeevi (2022), but instead applies more broadly to settings where distributions may evolve with the horizon.

### 4.2 $\varepsilon$-GREEDY WITH LCB THRESHOLDING

While the Explore-then-Decide method achieves a sharp competitive ratio, its deterministic separation between exploration and decision phases—and the fact that exploration is confined to the early stages—limits its practicality in applications where exploration spread across time is preferable, such as online advertising or sequential recommendation systems. To address this, we propose an $\varepsilon$-Greedy approach (Algorithm 2) that selects decision stages uniformly at random over the time horizon. The details of the strategy are described as follows.

**Randomized Exploration.** At each stage $i \in [n]$, we draw a Bernoulli random variable $b_i \sim$ Bernoulli$(\varepsilon)$, where $\varepsilon = \sqrt{\ell_n/n}$ with setting $\ell_n = o(n)$.

- If $b_i = 1$, we perform exploration by observing the noisy reward $y_i$ and feature $x_i$, and update, $\hat{\theta}_i = V_i^{-1} \sum_{t \in \mathcal{I}_i} y_t x_t$, where $V_i = \sum_{t \in \mathcal{I}_i} x_t x_t^\top + \beta I_d$ for a constant $\beta > 0$.
- If $b_i = 0$, we enter the decision phase and determine whether to stop based on a dynamic threshold.

Unlike the Explore-then-Decide method, here the exploration rounds are distributed over the entire horizon. Consequently, $\hat{\theta}_i$ and $V_i$ are updated continuously, which in turn affects both the LCB and the threshold dynamically, described below.

**Lower Confidence Bound.** We redefine the LCB for $X_i = x_i^\top \theta$ as

$$X_i^{\mathrm{LCB}} = x_i^\top \hat{\theta}_i - \xi_i(x_i), \tag{3}$$

where $\xi_i(x_i) := \sqrt{x_i^\top V_i^{-1} x_i} \left( \sigma \sqrt{d \log(n + n|\mathcal{I}_i|L/d\beta)} + \sqrt{S\beta} \right)$.

**Decision with LCB Threshold.** Using a CDF of $\mathbb{P}_{z \sim \mathcal{D}_x}(Z_i^{\mathrm{LCB}} \leq \alpha \mid \hat{\theta}_i, V_i, \mathcal{I}_i)$ where $Z_i^{\mathrm{LCB}} = z^\top \hat{\theta}_i - \xi_i(z)$, for each $i \in [n]$, we set the dynamic threshold $\alpha_i$ such that

$$\mathbb{P}_{z \sim \mathcal{D}_x}(Z_i^{\mathrm{LCB}} \leq \alpha_i \mid \hat{\theta}_i, V_i, \mathcal{I}_i) = 1 - \frac{1}{n}. \tag{4}$$

The algorithm stops at stage $i$ if $X_i^{\mathrm{LCB}} \geq \alpha_i$, in which case we set $\tau = i$. Unlike Explore-then-Decide, this procedure employs a dynamic threshold. Recall $\lambda = \lambda_{\min}(\mathbb{E}_{x \sim \mathcal{D}_x}[xx^\top])$. Then, the algorithm satisfies the following theorem (see Appendix A.3 for the proof).

**Theorem 4.4.** *The stopping policy $\tau$ of Algorithm 2 with $\varepsilon = \sqrt{\ell_n/n}$, $\ell_n = o(n)$, $\ell_n = \Omega(\max\{\frac{L \log d \log n}{\lambda}, \log n\})$, and a constant $\beta > 0$, achieves an asymptotic competitive ratio of*

$$\liminf_{n \to \infty} \frac{\mathbb{E}[X_\tau]}{\mathbb{E}[\max_{i \in [n]} X_i]} \geq 1 - \frac{1}{e} - \limsup_{n \to \infty} \varepsilon_n,$$

---

**Algorithm 2** $\varepsilon$-Greedy with LCB Thresholding ($\varepsilon$-Greedy-LCBT)

---

**Input:** Bernoulli parameter $\varepsilon$; regularization parameter $\beta$
**Output:** Stopping time $\tau$

1 **for** $i = 1, \ldots, n$ **do**
2      Sample $b_i \sim \text{Bernoulli}(\varepsilon)$
3      **if** $b_i = 1$ **then**
4          $\mathcal{I}_i \leftarrow \mathcal{I}_{i-1} \cup \{i\}$
5          Observe $(x_i, y_i)$
6          $V_i \leftarrow \sum_{t \in \mathcal{I}_i} x_t x_t^\top + \beta I_d$; $\hat{\theta}_i \leftarrow V_i^{-1} \sum_{t \in \mathcal{I}_i} y_t x_t$
7      **else**
8          $\mathcal{I}_i \leftarrow \mathcal{I}_{i-1}, \hat{\theta}_i \leftarrow \hat{\theta}_{i-1}, V_i \leftarrow V_{i-1}$
9          Observe $(x_i, y_i)$
10         Compute $X_i^{\text{LCB}}$ from (3) and $\alpha_i$ using (4)
11         **if** $X_i^{\text{LCB}} \geq \alpha_i$ **then**
12          Stop with $\tau \leftarrow i$

---

*where*

$$\varepsilon_n := \mathcal{O}\left( \frac{1}{\mathbb{E}\big[\max_{i \in [n]} X_i\big]} \sqrt{\frac{L(\sigma^2 d + S)\log(Ln)}{\lambda \ell_n}} \right).$$

*Furthermore, by setting $\ell_n = \frac{L(\sigma^2 d + S)}{\lambda} f(n) \log(Ln)$ for some function $f(n)$ (e.g., $f(n) = \Theta(\log^p n)$ for $p > 0$, or $\Theta(n^q)$ for $0 < q < 1$) satisfying $\ell_n = o(n)$, if $\text{OPT} = \omega(1/\sqrt{f(n)})$, then*

$$\liminf_{n \to \infty} \frac{\mathbb{E}[X_\tau]}{\mathbb{E}[\max_{i \in [n]} X_i]} \geq 1 - \frac{1}{e}.$$

Notably, the $\varepsilon$-Greedy approach achieves the same competitive ratio as established for the Explore-then-Decide method in Corollary 4.3, while ensuring uniformly random decision stages.

## 5 NON-IDENTICAL DISTRIBUTIONS

In this section, we consider the setting where the distributions $\mathcal{D}_i$ are not identical across $i \in [n]$. In what follows, we propose algorithms and analyze their competitive ratios.

### 5.1 EXPLORE-THEN-DECIDE WITH LCB THRESHOLDING

We build on the Explore-then-Decide framework in Algorithm 1, adapting the thresholding policy accordingly. In the initial exploration phase of length $\ell_n$, we collect data and estimate $\hat{\theta} = V^{-1} \sum_{t=1}^{\ell_n} y_t x_t$, where $V = \sum_{t=1}^{\ell_n} x_t x_t^\top + \beta I$ for a constant $\beta > 0$. In the subsequent decision phase, we apply LCB-based thresholding for non-identical distributions, as described below.

**Decision with LCB Threshold.** Let $\{z_s\}_{s=\ell_n+1}^n$ be independent random vectors with $z_s \sim \mathcal{D}_{x,s}$. We define the threshold:

$$\alpha := \frac{1}{2} \mathbb{E}\left[ \max_{s \in [\ell_n+1, n]} z_s^\top \hat{\theta} \mid \hat{\theta} \right]. \tag{5}$$

Recall the lower confidence bound for $X_i$ in the Explore-then-Decide framework: $X_i^{\text{LCB}} = x_i^\top \hat{\theta} - \xi(x_i)$, where $\xi(x_i) := \sqrt{x_i^\top V^{-1} x_i}(\sigma\sqrt{d \log(n + n\ell_n L/d\beta)} + \sqrt{S\beta})$. Then, the algorithm stops at stage $i$ if $X_i^{\text{LCB}} \geq \alpha$.

For notational convenience, let $\lambda' = \min_{i \in [n]} \lambda_{\min}\big(\mathbb{E}_{x \sim \mathcal{D}_{x,i}}[xx^\top]\big)$. Then, the algorithm satisfies with the following theorem (see Appendix A.5 for the proof).

**Theorem 5.1.** *Consider the stopping policy $\tau$ of Algorithm 1 with $\ell_n = o(n)$, $\ell_n = \omega(\frac{L \log d}{\lambda'})$, and a constant $\beta > 0$, where the threshold value is chosen according to* (5). *Then*

$$\liminf_{n \to \infty} \frac{\mathbb{E}[X_\tau]}{\mathbb{E}\big[\max_{i \in \{\ell_n+1,\dots,n\}} X_i\big]} \geq \frac{1}{2} - \limsup_{n \to \infty} \varepsilon_n,$$

*where*

$$\varepsilon_n := \mathcal{O}\left( \frac{1}{\mathbb{E}\big[\max_{i \in \{\ell_n+1,\dots,n\}} X_i\big]} \sqrt{\frac{L(\sigma^2 d + S) \log(Ln)}{\lambda' \ell_n}} \right).$$

*Let $\widetilde{\mathrm{OPT}} := \mathbb{E}\big[\max_{i \in \{\ell_n+1,\dots,n\}} X_i\big]$ and set $\ell_n = \frac{L(\sigma^2 d+S)}{\lambda'} f(n) \log(Ln)$ for some function $f(n)$ (e.g., $f(n) = \Theta(\log^p n)$ for $p > 0$, or $\Theta(n^q)$ for $0 < q < 1$) satisfying $\ell_n = o(n)$. If $\widetilde{\mathrm{OPT}} = \omega(1/\sqrt{f(n)})$, then*

$$\liminf_{n \to \infty} \frac{\mathbb{E}[X_\tau]}{\mathbb{E}[\max_{i \in [\ell_n+1,n]} X_i]} \geq \frac{1}{2}.$$

In the theorem, we target the relaxed prophet of $\mathbb{E}[\max_{i \in [\ell_n+1,n]} X_i]$ due to the inherent difficulty of the non-i.i.d. setting against the original prophet, as shown in Proposition 5.2 (see Appendix A.4 for the proof).

**Proposition 5.2.** *There exist non-identical distributions $\{\mathcal{D}_{x,i}\}_{i=1}^n$ for the feature vectors $x_i$'s, and a parameter vector $\theta \in \mathbb{R}^d$, such that when observing noise-free rewards $X_i = x_i^\top \theta$ for $i \in [n]$, the following holds: for any stopping rule $\tau$,*

$$\frac{\mathbb{E}[X_\tau]}{\mathbb{E}\big[\max_{i \in [n]} X_i\big]} \leq \min\left\{ \frac{1}{2}, \frac{1}{d} \right\}.$$

Proposition 5.2 shows that, even in the noise-free case ($\sigma = 0$), the initial stages must be sacrificed to learn $\theta$. For the standard prophet of $\mathbb{E}[\max_{i \in [n]} X_i]$, the competitive ratio approaches zero with large enough $d$ (e.g. $d = \log(n)$). This motivates our focus on the relaxed prophet benchmark $\mathbb{E}\big[\max_{i \in \{\ell_n+1,\dots,n\}} X_i\big]$, under which meaningful guarantees are attainable.

The optimal competitive ratio for non-identical distributions is known to be $1/2$ (Samuel-Cahn, 1984). In our setting with unknown distributions, Theorem 5.1 shows that attaining this ratio requires relaxing the prophet benchmark by excluding the initial exploration phase. In the next subsection, we present the case with a logarithmic number of offline samples under which the optimal prophet benchmark can be attained in our setting.

## 5.2 EXPLORE-THEN-DECIDE WITH WINDOW ACCESS

In the standard non-identical distribution setting, items are revealed sequentially, and the gambler must decide immediately whether to accept or reject the *current* observation. As discussed in Marshall et al. (2020); Benomar et al. (2024), this assumption, however, can be overly pessimistic: in many practical scenarios, early opportunities are not irrevocably lost but may remain available for a short period of time. For instance, in a hiring process, one may be able to interview several candidates sequentially before making a final choice among them.

**Window Access.** Motivated by this observation, we consider a mild relaxation of the standard setting by using window access for the previous time steps, same as Marshall et al. (2020). More specifically, given a window size $w_n$, at time $i$ the decision-maker has access to the indices in the window $[i - w_n + 1, i]$ and may choose any index in this window at which to stop; if no such index is selected, the process continues to time $i + 1$. Interestingly, even for any $w_n \leq n - 1$, the non-i.i.d. setting remains as hard as the standard model (corresponding to window size $w_n = 1$), in the sense that the optimal competitive ratio does not improve, as shown in the following (see Appendix A.6 for the proof).

**Proposition 5.3.** *In the non-i.i.d. setting with window access of size $w_n \leq n - 1$, for any algorithm, there always exist non-identical distributions such that the competitive ratio is bounded above by*

$$\frac{\mathbb{E}[X_\tau]}{\mathbb{E}[\max_{i \in [n]} X_i]} \leq 1/2.$$

These observations raise the following question: *Can the optimal competitive ratio under window access also be achieved in the setting of unknown non-identical distributions and noisy reward observations? If so, what window size $w_n$ is required, and how frequently is window access required?*

To handle this setting, we propose an algorithm (Algorithm 3 in Appendix A.7) adopting the Explore-then-Decide method. After the exploration phase, at time $\ell_n + 1$, the decision-maker is granted window access with size $w_n = \ell_n + 1$ and may stop at any index in $[\ell_n + 1]$; if no such index is selected, the process continues to time $\ell_n + 2$.

Notably, our algorithm requires only a single window access at time $\ell_n + 1$, with window size $\ell_n + 1$. Due to space constraints, we defer the detailed description of the algorithm to Appendix A.7. In what follows, we provide a theorem for the competitive ratio of the method with window access (see Appendix A.8 for the proof).

**Theorem 5.4.** *In the non-i.i.d. setting with unknown distributions and window access of size $w_n > \ell_n$, the stopping policy $\tau$ of Algorithm 3 with $\ell_n = o(n)$, $\ell_n = \omega(\frac{L \log d}{\lambda'})$, and a constant $\beta > 0$ achieves the following asymptotic competitive ratio:*

$$\liminf_{n \to \infty} \frac{\mathbb{E}[X_\tau]}{\mathbb{E}\big[\max_{i \in [n]} X_i\big]} \geq \frac{1}{2} - \limsup_{n \to \infty} \varepsilon_n,$$

*where*

$$\varepsilon_n := \mathcal{O}\left( \frac{1}{\mathbb{E}\big[\max_{i \in [n]} X_i\big]} \left( \sqrt{\frac{L(\sigma^2 d + S) \log(Ln)}{\lambda' \ell_n}} + \sqrt{LS}\left( \frac{1}{n} + \frac{d}{e^{\lambda' \ell_n / 8L}} \right) \right) \right).$$

*Furthermore, by setting $\ell_n = \frac{L(\sigma^2 d + S)}{\lambda'} f(n) \log(Ln)$ for some function $f(n)$ (e.g., $f(n) = \Theta(\log^p n)$ for $p > 0$, or $\Theta(n^q)$ for $0 < q < 1$) satisfying $\ell_n = o(n)$, if $\mathrm{OPT} = \omega(\max\{1/\sqrt{f(n)}, \sqrt{LS}/n\})$, then*

$$\liminf_{n \to \infty} \frac{\mathbb{E}[X_\tau]}{\mathbb{E}[\max_{i \in [n]} X_i]} \geq \frac{1}{2}.$$

Notably, Algorithm 3 achieves the optimal ratio, matching the upper bound in Proposition 5.3.

**Remark 5.5.** *The $1/2$ guarantee can also be obtained by running LCB-Thresholding with $\ell_n$ offline samples, without window access. Exploiting the linear structure allows our method to achieve the optimal $1/2$ ratio using only $\ell_n$ samples—for example, $\ell_n = O(\log^{p+1} n)$ when $f(n) = \Theta(\log^p n)$ for $p > 0$. Further details are provided in Appendix A.9.*

## 6 EXPERIMENTS

We evaluate our algorithms on synthetic data with Gaussian reward noise of variance $\sigma^2$. We set $d = 2$, $\ell_n = n^{2/3}$, and $\beta = 1$. As no prior method directly fits our setting, we use the `Gusein-Zade` rule (Gusein-Zade, 1966) as a baseline: it skips the first $n/e$ samples and then stops at the first record exceeding the maximum of this prefix. This rule guarantees a worst-case competitive ratio of $1/e$ for unknown i.i.d. distributions (Correa et al., 2019a) and can be asymptotically optimal for certain distributions (Goldenshluger & Zeevi, 2022). All results are averaged over independent Monte Carlo runs.[1]

**I.i.d. distribution.** In the i.i.d. setting with $n = 100{,}000$, we draw $\theta$ and $x_i$ coordinate-wise uniformly and then normalize. Figure 1 shows that our algorithms, `ETD-LCBT(iid)` (Algorithm 1) and `ε-Greedy-LCBT` (Algorithm 2), achieve competitive ratios exceeding $1 - 1/e$, consistent with Corollary 4.3 and Theorem 4.4, and outperform `Gusein-Zade`. Moreover, the performance gap widens as $\sigma^2$ increases, suggesting that our approaches are robust to observation noise.

---

[1]Code: https://github.com/junghunkim7786/LearningProphetInequalityNoisyObservation

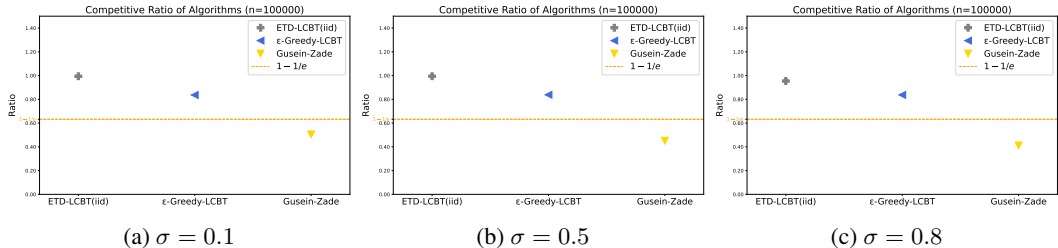

Figure 1: Competitive ratio under the i.i.d. setting with noise standard deviation $\sigma$.

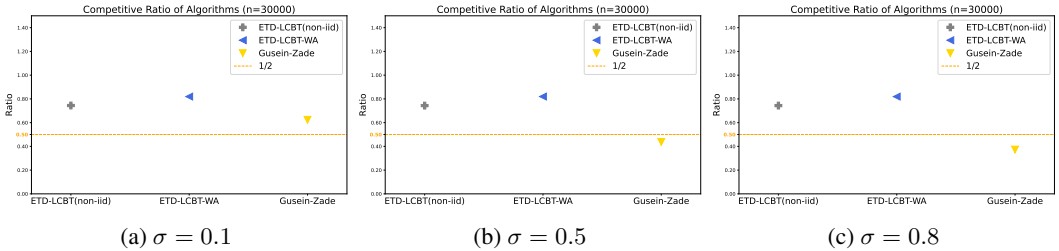

Figure 2: Competitive ratio under non-identical distributions with noise standard deviation $\sigma$.

**Non-i.i.d. distributions.** For $n = 30,000$, we draw $\theta$ as above, while each $x_i$ comes from a distinct distribution by first sampling a coordinate-wise range and then drawing $x_i$ uniformly within that range. Figure 2 shows that the window-access method `ETD-LCBT-WA` (Algorithm 3) achieves a competitive ratio above $1/2$, matching Theorem 5.4. Even `ETD-LCBT(non-iid)` (Algorithm 1), analyzed only against a relaxed benchmark (Theorem 5.1), empirically exceeds $1/2$ against the standard prophet. As expected, `ETD-LCBT-WA` improves upon `ETD-LCBT(non-iid)`, and both outperform `Gusein-Zade`, with a gap that widens as $\sigma^2$ increases.

## 7 CONCLUSION

We introduced a new framework for prophet inequalities under noisy observations and unknown reward distributions, motivated by real-world applications where noisy reward and contextual information are observable but reward distributions are not. By combining learning with LCB-based stopping rules, we achieved the sharp competitive ratio of $1 - 1/e$ in the i.i.d. setting. For non-identical distributions, we showed that the optimal bound of $1/2$ can be attained under window access or under a relaxed prophet benchmark. Our empirical results demonstrate the efficiency of our algorithms.

## REPRODUCIBILITY STATEMENT

All theoretical claims are stated under explicit assumptions and are supported by complete proofs in the appendix. Algorithmic details, including full pseudocode, are provided in the main paper and appendix. For the experimental results, we describe the data-generation process in the main paper and release the complete source code needed to reproduce all figures and numerical results at the provided URL.

## ACKNOWLEDGEMENTS

The authors acknowledge the support of ANR through the PEPR IA FOUNDRY project (ANR-23-PEIA-0003) and the Doom project (ANR-23-CE23-0002), as well as the ERC through the Ocean project (ERC-2022-SYG-OCEAN-101071601).

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

# A  APPENDIX

## A.1  PROOF OF PROPOSITION 4.1

We give a construction under additive Gaussian noise, which is $\sigma$-sub-Gaussian.

Let $X_1, \ldots, X_n$ be i.i.d. Bernoulli with success probability $p_n = c/n$ for some fixed $c \in (0, \infty)$. Let $(\eta_i)_{i=1}^n$ be i.i.d. $\mathcal{N}(0, \sigma^2)$ for constant $\sigma > 0$, independent of $(X_i)_{i=1}^n$, and suppose the gambler observes

$$Y_i = X_i + \eta_i, \qquad i = 1, \ldots, n.$$

Let $\tau$ be any (possibly randomized) index valued in $\{1, \ldots, n\}$ that is measurable with respect to $Y = (Y_1, \ldots, Y_n)$. Then, conditioning on $Y$ and using $\sum_{i=1}^{n+1} \Pr(\tau = i \mid Y) = 1$ and $X_{n+1} := 0$,

$$\mathbb{E}[X_\tau \mid Y] = \sum_{i=1}^{n+1} \Pr(\tau = i \mid Y)\, \mathbb{E}[X_i \mid Y] \leq \max_{1 \leq i \leq n} \mathbb{E}[X_i \mid Y].$$

Moreover, since $(X_i, \eta_i)$ are independent across $i$, we have $\mathbb{E}[X_i \mid Y] = \mathbb{E}[X_i \mid Y_i] = \Pr(X_i = 1 \mid Y_i)$. Let

$$\pi(y) := \Pr(X_i = 1 \mid Y_i = y).$$

By Bayes' rule for the Gaussian likelihoods,

$$\pi(y) = \frac{p_n \varphi_\sigma(y - 1)}{(1 - p_n)\varphi_\sigma(y) + p_n \varphi_\sigma(y - 1)}, \quad \text{where } \varphi_\sigma(t) = \frac{1}{\sqrt{2\pi}\sigma} e^{-t^2/(2\sigma^2)}.$$

Hence, using $(1 - p_n)\varphi_\sigma(y)$ as a lower bound on the denominator,

$$\pi(y) \leq \frac{p_n}{1 - p_n} \cdot \frac{\varphi_\sigma(y - 1)}{\varphi_\sigma(y)} = \frac{p_n}{1 - p_n} \exp\left(\frac{2y - 1}{2\sigma^2}\right).$$

Now define the event

$$\mathcal{E}_n := \left\{ \max_{1 \leq i \leq n} Y_i \leq t_n \right\}, \qquad t_n := 1 + \sigma\sqrt{8 \log n}.$$

Since $X_i \in \{0, 1\}$, $Y_i > t_n$ implies $\eta_i > t_n - 1$, and Gaussian tails give

$$\Pr(\mathcal{E}_n^c) \leq n \Pr(\eta_1 > t_n - 1) \leq n \exp\left(-\frac{(t_n - 1)^2}{2\sigma^2}\right) = n e^{-4\log n} = n^{-3} \xrightarrow[n \to \infty]{} 0.$$

On $\mathcal{E}_n$, we have $\max_i \pi(Y_i) \leq \pi(t_n)$, so

$$\mathbb{E}[X_\tau] \leq \mathbb{E}\left[\max_{1 \leq i \leq n} \pi(Y_i)\right] \leq \pi(t_n) + \Pr(\mathcal{E}_n^c) \leq \frac{2c}{n} \exp\left(\frac{2t_n - 1}{2\sigma^2}\right) + o(1) \xrightarrow[n \to \infty]{} 0,$$

where we used $p_n = c/n$ and $1/(1 - p_n) \leq 2$ for large $n$.

On the other hand, the oracle that sees the true $X$'s obtains $\max_i X_i$, so

$$\mathbb{E}\left[\max_{1 \leq i \leq n} X_i\right] = 1 - (1 - p_n)^n = 1 - \left(1 - \frac{c}{n}\right)^n \xrightarrow[n \to \infty]{} 1 - e^{-c} > 0.$$

Therefore,

$$\frac{\mathbb{E}[X_\tau]}{\mathbb{E}[\max_i X_i]} \xrightarrow[n \to \infty]{} 0,$$

and the conclusion holds for any algorithm.

## A.2   PROOF OF THEOREM 4.2

We first provide a lemma for estimation error.

**Lemma A.1** (Theorem 2 in Abbasi-Yadkori et al. (2011))**.** *For $\delta > 0$, we have*

$$\mathbb{P}\left(\|\hat{\theta} - \theta\|_V \leq \sqrt{S\beta} + \sigma\sqrt{d\log\left(\frac{1 + \ell_n L/d\beta}{\delta}\right)}\right) \geq 1 - \delta.$$

The above lemma implies that

$$\mathbb{P}\left(\left|x^\top(\hat{\theta} - \theta)\right| \leq \sqrt{x^\top V^{-1}x}\left(\sigma\sqrt{d\log(n + n\ell_n L/d\beta)} + \sqrt{S\beta}\right), \forall x \in \mathbb{R}^d\right) \geq 1 - 1/n. \quad (6)$$

We define an event $\mathcal{E}_1 = \{|x^\top(\hat{\theta} - \theta)| \leq \sqrt{x^\top V^{-1}x}(\sigma\sqrt{d\log(n + n\ell_n L/d\beta)} + \sqrt{S\beta}), \forall x \in \mathbb{R}^d\}$, which holds with $\mathbb{P}(\mathcal{E}_1) \geq 1 - \frac{1}{n}$.

We define $g := \sqrt{L\|V^{-1}\|_2}(\sigma\sqrt{d\log(n + n\ell_n L/d\beta)} + \sqrt{S\beta})$. Recall

$$\xi(x_i) = \sqrt{x_i^\top V^{-1}x_i}(\sigma\sqrt{d\log(n + n\ell_n L/d\beta)} + \sqrt{S\beta}).$$

Then we have $\xi(x_i) \leq g$. Under $\mathcal{E}_1$, for any $i > \ell_n$ we have

$$X_i - 2g \leq x_i^\top\hat{\theta} - g \leq x_i^\top\hat{\theta} - \xi(x_i) = X_i^{\text{LCB}} \leq X_i. \quad (7)$$

We define $\alpha^*$ s.t. $\mathbb{P}_{X \sim \mathcal{D}}(X \leq \alpha^*) = 1 - \frac{1}{n}$. Then we have the following lemma regarding the bounds for the threshold value. We denote $\mathcal{H}_{\ell_n} = \{\hat{\theta}, V\}$.

**Lemma A.2.** *Under $\mathcal{E}_1$, for any $\mathcal{H}_{\ell_n}$, we have*

$$\alpha^* - 2g \leq \alpha \leq \alpha^*.$$

*Proof.* For $z \sim \mathcal{D}_x$, we define $Z = z^\top\theta$ and $\hat{Z} = z^\top\hat{\theta}$. Then, under $\mathcal{E}_1$, for any given $\mathcal{H}_{\ell_n}$, we have $Z - 2g \leq \hat{Z} - g \leq \hat{Z} - \xi(z) \leq Z$ with $\xi(z) \leq g$. Since $\hat{Z} - \xi(z) \leq Z$ and $\mathbb{P}(\hat{Z} - \xi(z) \geq \alpha \mid \mathcal{H}_{\ell_n}) = \mathbb{P}(Z \geq \alpha^* \mid \mathcal{H}_{\ell_n}) = 1/n$, we can easily obtain

$$\alpha \leq \alpha^*.$$

Likewise, for $\alpha'$ s.t. $\mathbb{P}(Z - 2g \geq \alpha' \mid \mathcal{H}_{\ell_n}) = \frac{1}{n}$, from $Z - 2g \leq \hat{Z} - \xi(z)$ and $\mathbb{P}(Z - 2g \geq \alpha' \mid \mathcal{H}_{\ell_n}) = \mathbb{P}(\hat{Z} - \xi(z) \geq \alpha \mid \mathcal{H}_{\ell_n}) = 1/n$, we have $\alpha' \leq \alpha$. Therefore, with $\alpha' + 2g = \alpha^*$ from $\mathbb{P}(Z \geq \alpha^* \mid \mathcal{H}_{\ell_n}) = \mathbb{P}(Z - 2g \geq \alpha' \mid \mathcal{H}_{\ell_n}) = 1/n$, we have

$$\alpha^* - 2g \leq \alpha,$$

which concludes the proof. $\square$

**Lemma A.3.** *For $l \geq 1$, let $z_1, \ldots, z_l \overset{i.i.d.}{\sim} \mathcal{D}_x$ satisfying Assumption 3.2. Recall $\lambda = \lambda_{\min}(\mathbb{E}_{z \sim \mathcal{D}_x}[zz^\top]) > 0$. Then*

$$\mathbb{P}\left(\frac{1}{l}\sum_{s=1}^l z_s z_s^\top \succeq \frac{\lambda}{2}I_d\right) \geq 1 - d\exp\left(-\frac{\lambda l}{8L}\right).$$

*Proof.* Let $\mu_{\min} = \lambda_{\min}(\mathbb{E}[\sum_{s=1}^l z_s z_s^\top])$. By the matrix Chernoff bound (Theorem 5.1.1 in Tropp et al. (2015)) for sums of independent PSD matrices with Assumption 3.2, for any $\delta \in [0, 1]$,

$$\Pr\left[\lambda_{\min}\left(\sum_{s=1}^l z_s z_s^\top\right) \leq (1 - \delta)\mu_{\min}\right] \leq d\left(\frac{e^{-\delta}}{(1-\delta)^{1-\delta}}\right)^{\mu_{\min}/L} \leq d\exp\left(-\frac{\delta^2}{2} \cdot \frac{\mu_{\min}}{L}\right).$$

Choosing $\delta = \frac{1}{2}$ yields

$$\Pr\left[\lambda_{\min}\left(\sum_{s=1}^l z_s z_s^\top\right) \leq \frac{\mu_{\min}}{2}\right] \leq d\exp\left(-\frac{\mu_{\min}}{8L}\right) = d\exp\left(-\frac{l\lambda}{8L}\right).$$

Equivalently, with probability at least $1 - d\exp(-\lambda l/(8L))$,

$$\sum_{s=1}^{l} z_s z_s^\top \succeq \frac{\mu_{\min}}{2} I_d = \frac{l\lambda}{2} I_d,$$

which completes the proof. $\qquad\square$

Let $\mathcal{E}_2 = \{\sum_{s=1}^{\ell_n} x_s x_s^\top \succeq \frac{\lambda\ell_n}{2} I_d\}$, which holds with probability at least $1 - \frac{d}{e^{\lambda\ell_n/8L}}$ from Lemma A.3. Then under $\mathcal{E}_2$, we have $\|V^{-1}\|_2 \le \|(\sum_{s=1}^{\ell_n} x_s x_s^\top)^{-1}\|_2 \le 2\frac{1}{\lambda\ell_n}$. Then, we have

$$\xi(x_i) \le \sqrt{L\|V^{-1}\|_2}(\sigma\sqrt{d\log(n + n\ell_n L/d\beta)} + S\sqrt{\beta})(= g)$$
$$\le \sqrt{L\frac{2}{\lambda\ell_n}}(\sigma\sqrt{d\log(n + n^2 L/d\beta)} + S\sqrt{\beta}).$$

Here we define $h := \sqrt{L\frac{2}{\lambda\ell_n}}(\sigma\sqrt{d\log(n + n^2 L/d\beta)} + S\sqrt{\beta})$ and $\mathcal{E} := \mathcal{E}_1 \cap \mathcal{E}_2$. For analyzing $X_\tau$, we first examine the probability that the stopping time $\tau$ equals $i$.

**Lemma A.4.** *For $i > \ell_n$, we have*

$$\mathbb{P}(\tau = i \mid \mathcal{H}_{\ell_n}) = \left(1 - \frac{1}{n}\right)^{i-\ell_n-1} \frac{1}{n}.$$

*Proof.* For $i > \ell_n$, we have

$$\begin{aligned}
&\mathbb{P}(\tau = i \mid \mathcal{H}_{\ell_n}) \\
&= \mathbb{P}(X_{\ell_n+1}^{\mathrm{LCB}} \le \alpha, \ldots, X_{i-1}^{\mathrm{LCB}} \le \alpha, X_i^{\mathrm{LCB}} > \alpha \mid \mathcal{H}_{\ell_n}) \\
&= \mathbb{P}(X_{\ell_n+1}^{\mathrm{LCB}} \le \alpha, \ldots, X_{i-1}^{\mathrm{LCB}} \le \alpha \mid \mathcal{H}_{\ell_n})\mathbb{P}(X_i^{\mathrm{LCB}} > \alpha \mid \mathcal{H}_{\ell_n}) \\
&= \mathbb{P}(X_{\ell_n+1}^{\mathrm{LCB}} \le \alpha, \ldots, X_{i-1}^{\mathrm{LCB}} \le \alpha \mid \mathcal{H}_{\ell_n})\frac{1}{n},
\end{aligned}$$

where the second equality is obtained from the fact that, given $\hat{\theta}$ and $V$, $X_i^{\mathrm{LCB}}$ is independent to $X_{\ell_n+1}^{\mathrm{LCB}}, \ldots, X_{i-1}^{\mathrm{LCB}}$. Similarly, for the last term above, we have

$$\begin{aligned}
&\mathbb{P}(X_{\ell_n+1}^{\mathrm{LCB}} \le \alpha, \ldots, X_{i-1}^{\mathrm{LCB}} \le \alpha \mid \mathcal{H}_{\ell_n})\frac{1}{n} \\
&= \mathbb{P}(X_{\ell_n+1}^{\mathrm{LCB}} \le \alpha, \ldots, X_{i-2}^{\mathrm{LCB}} \le \alpha \mid \mathcal{H}_{\ell_n})\mathbb{P}(X_{i-1}^{\mathrm{LCB}} \le \alpha \mid \mathcal{H}_{\ell_n})\frac{1}{n} \\
&= \mathbb{P}(X_{\ell_n+1}^{\mathrm{LCB}} \le \alpha, \ldots, X_{i-2}^{\mathrm{LCB}} \le \alpha \mid \mathcal{H}_{\ell_n})\left(1 - \frac{1}{n}\right)\frac{1}{n} \\
&\quad\vdots \\
&= \left(1 - \frac{1}{n}\right)^{i-\ell_n-1}\frac{1}{n},
\end{aligned}$$

which concludes the proof. $\qquad\square$

From the exploration phase in the algorithm, we have $\mathbb{P}(\tau = i \mid \mathcal{E}) = 0$ for all $1 \leq i \leq \ell_n$. Therefore, we have

$$
\mathbb{E}\left[\mathbb{E}[X_\tau \mathbb{1}(\mathcal{E}) \mid \mathcal{H}_{\ell_n}]\right]
$$

$$
= \mathbb{E}\left[\sum_{i=1}^{n} \mathbb{P}(\tau = i \mid \mathcal{H}_{\ell_n})\mathbb{E}[X_i \mathbb{1}(\mathcal{E}) \mid \tau = i, \mathcal{H}_{\ell_n}]\right]
$$

$$
= \mathbb{E}\left[\sum_{i=\ell_n+1}^{n} \mathbb{P}(\tau = i \mid \mathcal{H}_{\ell_n})\mathbb{E}[X_i \mathbb{1}(\mathcal{E}) \mid \tau = i, \mathcal{H}_{\ell_n}]\right]
$$

$$
\geq \mathbb{E}\left[\sum_{i=\ell_n+1}^{n} \mathbb{P}(\tau = i \mid \mathcal{H}_{\ell_n})\mathbb{E}[X_i^{\mathrm{LCB}} \mathbb{1}(\mathcal{E}) \mid \tau = i, \mathcal{H}_{\ell_n}]\right]
$$

$$
\geq \mathbb{E}\left[\sum_{i=\ell_n+1}^{n} \left(1 - \frac{1}{n}\right)^{i-\ell_n-1} \frac{1}{n}\mathbb{E}[X_i^{\mathrm{LCB}} \mathbb{1}(\mathcal{E}) \mid \tau = i, \mathcal{H}_{\ell_n}]\right]
$$

$$
= \mathbb{E}\left[\sum_{i=\ell_n+1}^{n} \left(1 - \frac{1}{n}\right)^{i-\ell_n-1} \frac{1}{n} \times \left(\mathbb{E}[\alpha \mathbb{1}(\mathcal{E}) \mid \tau = i, \mathcal{H}_{\ell_n}] + \mathbb{E}[(X_i^{\mathrm{LCB}} - \alpha)\mathbb{1}(\mathcal{E}) \mid X_i^{\mathrm{LCB}} \geq \alpha, \mathcal{H}_{\ell_n}]\right)\right]
$$

$$
= \mathbb{E}\left[\sum_{i=\ell_n+1}^{n} \left(1 - \frac{1}{n}\right)^{i-\ell_n-1} \frac{1}{n} \times \left(\mathbb{E}[\alpha \mathbb{1}(\mathcal{E}) \mid \mathcal{H}_{\ell_n}] + \mathbb{E}[(X_i^{\mathrm{LCB}} - \alpha)^+\mathbb{1}(\mathcal{E}) \mid X_i^{\mathrm{LCB}} \geq \alpha, \mathcal{H}_{\ell_n}]\right)\right]
$$

$$
= \mathbb{E}\left[\sum_{i=\ell_n+1}^{n} \left(1 - \frac{1}{n}\right)^{i-\ell_n-1} \frac{1}{n} \times \left(\mathbb{E}[\alpha \mathbb{1}(\mathcal{E}) \mid \mathcal{H}_{\ell_n}] + \frac{\mathbb{E}[(X_i^{\mathrm{LCB}} - \alpha)^+\mathbb{1}(\mathcal{E}) \mid \mathcal{H}_{\ell_n}]}{\mathbb{P}(X_i^{\mathrm{LCB}} \geq \alpha \mid \mathcal{H}_{\ell_n})}\right)\right]
$$

$$
\geq \mathbb{E}\left[\sum_{i=\ell_n+1}^{n} \left(1 - \frac{1}{n}\right)^{i-\ell_n-1} \frac{1}{n} \times \left(\mathbb{E}[\alpha^* \mathbb{1}(\mathcal{E}) - 2g\mathbb{1}(\mathcal{E}) \mid \mathcal{H}_{\ell_n}] + n\mathbb{E}\left[(X_i - 2g - \alpha^*)^+ \mathbb{1}(\mathcal{E}) \mid \mathcal{H}_{\ell_n}\right]\right)\right],
$$

$$
\tag{8}
$$

where the second inequality is obtained from Lemma A.4, and the last inequality is obtained from Lemma A.2 and (7). Let i.i.d. $Z_s \sim \mathcal{D}$ for $s \in [n]$. Then, with $\ell_n = o(n)$, for the last term in (8), we have

$$
\mathbb{E}\left[\sum_{i=\ell_n+1}^{n} \left(1 - \frac{1}{n}\right)^{i-\ell_n-1} \frac{1}{n} \times \left(\mathbb{E}[(\alpha^* - 2g)\mathbb{1}(\mathcal{E}) \mid \mathcal{H}_{\ell_n}] + n\mathbb{E}\left[(X_i - 2g - \alpha^*)^+ \mathbb{1}(\mathcal{E}) \mid \mathcal{H}_{\ell_n}\right]\right)\right]
$$

$$
\geq \sum_{i=\ell_n+1}^{n} \left(1 - \frac{1}{n}\right)^{i-\ell_n-1} \frac{1}{n} \times \mathbb{E}\left[\mathbb{E}[\alpha^*\mathbb{1}(\mathcal{E}) \mid \mathcal{H}_{\ell_n}] - 2\mathbb{E}[h\mathbb{1}(\mathcal{E}) \mid \mathcal{H}_{\ell_n}] + n\mathbb{E}[(X_i - 2h - \alpha^*)^+ \mathbb{1}(\mathcal{E}) \mid \mathcal{H}_{\ell_n}]\right]
$$

$$
= \sum_{i=\ell_n+1}^{n} \left(1 - \frac{1}{n}\right)^{i-\ell_n-1} \frac{1}{n} \times \mathbb{E}\left[\mathbb{E}[\alpha^*\mathbb{1}(\mathcal{E}) \mid \mathcal{H}_{\ell_n}] - 2\mathbb{E}[h\mathbb{1}(\mathcal{E}) \mid \mathcal{H}_{\ell_n}] + \mathbb{E}\left[\sum_{s\in[n]}(Z_s - 2h - \alpha^*)^+ \mathbb{1}(\mathcal{E}) \mid \mathcal{H}_{\ell_n}\right]\right]
$$

$$
\geq \sum_{i=\ell_n+1}^{n} \left(1 - \frac{1}{n}\right)^{i-\ell_n-1} \frac{1}{n}\mathbb{E}\left[\mathbb{E}[\alpha^*\mathbb{1}(\mathcal{E}) \mid \mathcal{H}_{\ell_n}] - 2\mathbb{E}[h\mathbb{1}(\mathcal{E}) \mid \mathcal{H}_{\ell_n}] + \mathbb{E}\left[\max_{s\in[n]}(Z_s - 2h - \alpha^*)^+ \mathbb{1}(\mathcal{E}) \mid \mathcal{H}_{\ell_n}\right]\right]
$$

$$
\geq \sum_{i=\ell_n+1}^{n} \left(1 - \frac{1}{n}\right)^{i-\ell_n-1} \frac{1}{n} \times \left(\alpha^*\mathbb{P}(\mathcal{E}) - 2h\mathbb{P}(\mathcal{E}) + \mathbb{E}\left[\max_{s\in[n]}(Z_s - 2h - \alpha^*) \mathbb{1}(\mathcal{E})\right]\right)
$$

$$= \sum_{i=\ell_n+1}^{n} \left(1 - \frac{1}{n}\right)^{i-\ell_n-1} \frac{1}{n} \times \left(\alpha^* \mathbb{P}(\mathcal{E}) - 2h\mathbb{P}(\mathcal{E}) + \mathbb{E}\left[\max_{s\in[n]} Z_s\right] \mathbb{P}(\mathcal{E}) - (2h+\alpha^*)\mathbb{P}(\mathcal{E})\right)$$

$$\geq \sum_{i=\ell_n+1}^{n} \left(1 - \frac{1}{n}\right)^{i-\ell_n-1} \frac{1}{n} \times \left(\mathbb{E}\left[\max_{s\in[n]} X_s\right] \mathbb{P}(\mathcal{E}) - 4h\right)$$

$$\geq \sum_{i=\ell_n+1}^{n} \left(1 - \frac{1}{n}\right)^{i-\ell_n-1} \frac{1}{n} \times \left(\mathbb{E}\left[\max_{s\in[n]} X_s\right] \mathbb{P}(\mathcal{E}) - O\left(\sqrt{\frac{L\log(nL)}{\lambda\ell_n}}(\sigma\sqrt{d} + \sqrt{S})\right)\right)$$

$$\geq \frac{1 - (1 - \frac{1}{n})^{n-\ell_n}}{1/n} \frac{1}{n} \left(\mathbb{E}\left[\max_{s\in[n]} X_s\right]\left(1 - \frac{1}{n} - \frac{d}{e^{\lambda\ell_n/8L}}\right) - O\left(\sqrt{\frac{L\log(nL)}{\lambda\ell_n}}(\sigma\sqrt{d} + \sqrt{S})\right)\right),$$

$$\tag{9}$$

where the first inequality is obtained from $g \leq h$.

Finally, from (8) and (9), we have

$$\liminf_{n\to\infty} \frac{\mathbb{E}[X_\tau]}{\mathbb{E}[\max_{i\in[n]} X_i]} = \liminf_{n\to\infty} \frac{\mathbb{E}\left[\mathbb{E}[X_\tau \mid \mathcal{H}_{\ell_n}]\right]}{\mathbb{E}[\max_{i\in[n]} X_i]}$$

$$\geq \liminf_{n\to\infty} \frac{\mathbb{E}\left[\mathbb{E}[X_\tau \mathbb{1}(\mathcal{E}) \mid \mathcal{H}_{\ell_n}]\right]}{\mathbb{E}[\max_{i\in[n]} X_i]}$$

$$\geq \liminf_{n\to\infty} \frac{1 - (1 - \frac{1}{n})^{n-\ell_n}}{1/n} \frac{1}{n} \left(\left(1 - \frac{1}{n} - \frac{d}{e^{\lambda\ell_n/8L}}\right) - \mathcal{O}\left(\frac{1}{\mathbb{E}[\max_{i\in[n]} X_i]} \sqrt{L\frac{\log(Ln)}{\lambda\ell_n}}(\sigma\sqrt{d} + \sqrt{S})\right)\right)$$

$$= \left(1 - \frac{1}{e}\right) - \mathcal{O}\left(\limsup_{n\to\infty} \frac{1}{\mathbb{E}[\max_{i\in[n]} X_i]} \sqrt{\frac{L(\sigma^2 d + S)\log(Ln)}{\lambda\ell_n}}\right),$$

where the last inequality is obtained from limits $\lim_{n\to\infty}(1 - 1/n)^n = 1/e$ and $\lim_{n\to\infty}(1 - 1/n)^{\ell_n} = 1$ (since $\ell_n = o(n)$) and $\ell_n = \omega(\frac{L\log d}{\lambda})$.

## A.3 PROOF OF THEOREM 4.4

**Lemma A.5** (Theorem 2 in Abbasi-Yadkori et al. (2011)). *We have*

$$\mathbb{P}\left(\forall i \in [n], \|\hat{\theta}_i - \theta\|_{V_i} \leq \sqrt{S\beta} + \sigma\sqrt{d\log\left(\frac{1 + |\mathcal{I}_i|L/d\beta}{\delta}\right)}\right) \geq 1 - \delta$$

The above lemma implies that

$$\mathbb{P}\left(\left|x^\top(\hat{\theta}_i - \theta)\right| \leq \sqrt{x^\top V_i^{-1} x}\left(\sigma\sqrt{d\log\left(n + n|\mathcal{I}_i|L/d\beta\right)} + \sqrt{S\beta}\right), \forall x \in \mathbb{R}^d, \forall i \in [n]\right) \geq 1 - 1/n.$$

$$\tag{10}$$

Let $a_n = \lceil\sqrt{n\ell_n}\rceil$. We define an event $\mathcal{E}_1 = \{|x^\top(\hat{\theta}_i - \theta)| \leq \sqrt{x^\top V_i^{-1} x}(\sigma\sqrt{d\log(n + n|\mathcal{I}_i|L/d\beta)} + \sqrt{S\beta}), \forall x \in \mathbb{R}^d, \forall i \in [a_n + 1, n]\}$. From (10), we have $\mathbb{P}(\mathcal{E}_1) \geq 1 - \frac{1}{n}$. Then we define $g := \sqrt{L\|V_{a_n}^{-1}\|_2}(\sigma\sqrt{d\log(n + n^2 L/d\beta)} + \sqrt{S\beta})$ so that, for $i > a_n$, $\xi_i(x_i) \leq g$ (recall $\xi_i(x_i) = \sqrt{x_i^\top V_i^{-1} x_i}(\sigma\sqrt{d\log(n + n|\mathcal{I}_i|L/d\beta)} + \sqrt{S\beta})$). We denote $\mathcal{H}_i = \{\hat{\theta}_i, V_i, \mathcal{I}_i\}$.

**Lemma A.6.** *Under $\mathcal{E}_1$, for any $i > a_n$, and any given $\mathcal{H}_i$, we have*

$$\alpha^* - 2g \leq \alpha_i \leq \alpha^*.$$

*Proof.* For $z \sim \mathcal{D}$, we define $Z = z^\top \theta$ and $\hat{Z}_i = z^\top \hat{\theta}_i$. Then, under $\mathcal{E}_1$, for any given $V_i$ and $\hat{\theta}_i$, we have $Z - 2g \le \hat{Z}_i - g \le \hat{Z}_i - \xi_i(z) \le Z$ with $\xi_i(z) \le g$.

Let $\alpha^*$ be the oracle threshold satisfying $\mathbb{P}(Z \ge \alpha^* | \mathcal{H}_i)(= \mathbb{P}(Z \ge \alpha^*)) = 1/n$. From $\hat{Z}_i - \xi_i(z) \le Z$ and $\mathbb{P}(\hat{Z}_i - \xi_i(z) \ge \alpha_i \mid \mathcal{H}_i) = \mathbb{P}(Z \ge \alpha^* \mid \mathcal{H}_i)(= 1/n)$, we can easily obtain

$$\alpha_i \le \alpha^*.$$

Likewise, for $\alpha'$ s.t. $\mathbb{P}(Z - 2g \ge \alpha' \mid \mathcal{H}_i) = \frac{1}{n}$, from $Z - 2g \le \hat{Z}_i - g \le \hat{Z}_i - \xi_i(z)$ and $\mathbb{P}(Z - 2g \ge \alpha' \mid \mathcal{H}_i) = \mathbb{P}(\hat{Z}_i - \xi_i(z) \ge \alpha_i \mid \mathcal{H}_i)$, we have $\alpha' \le \alpha_i$. Therefore, with $\alpha' + 2g = \alpha^*$ from $\mathbb{P}(Z - 2g \ge \alpha' \mid \mathcal{H}_i) = \mathbb{P}(Z \ge \alpha^* \mid \mathcal{H}_i) = 1/n$, we have

$$\alpha^* - 2g \le \alpha_i,$$

which concludes the proof. $\qquad\square$

**Lemma A.7** (Multiplicative Chernoff Bound)**.** *Let $Z_1, \ldots Z_l$ be Bernoulli random variables with mean $\mu$. Then for $0 \le \delta \le 1$ we have*

$$\mathbb{P}\left( \left| \sum_{s=1}^{l} Z_s - l\mu \right| \ge \delta l \mu \right) \le 2\exp(-\delta^2 l\mu/3)$$

From the above lemma, we define $\mathcal{E}_2 = \left\{ \left| |\mathcal{I}_i| - i\sqrt{\ell_n/n} \right| \le \frac{1}{2} i \sqrt{\ell_n/n}, i \in \{a_n, n\} \right\}$, which holds with probability at least $1 - 2\exp(\frac{-\ell_n}{12}) - 2\exp(\frac{-\sqrt{n\ell_n}}{12})$.

From Lemma A.3, for any $l \ge 1$, suppose $z_1, \ldots, z_l \sim \mathcal{D}_x$ are i.i.d drawn from a distribution $\mathcal{D}_x$ satisfying Assumption 3.2. Recall $\lambda = \lambda_{\min}(\mathbb{E}_{z \sim \mathcal{D}_x}[zz^\top]) > 0$. We have that

$$\mathbb{P}\left( \frac{1}{l} \sum_{s=1}^{l} z_s z_s^\top \succeq \frac{\lambda}{2} I_d \right) \ge 1 - d\exp\left( -\frac{\lambda l}{8L} \right).$$

Then, we define $\mathcal{E}_3 = \{ \sum_{s \in \mathcal{I}_{a_n}} x_s x_s^\top \succeq \frac{|\mathcal{I}_{a_n}|}{2} \lambda I_d \}$, which holds, under $\mathcal{E}_2$, with probability at least $1 - \frac{d}{e^{\lambda \ell_n/(16L)}}$. This implies $\mathbb{P}(\mathcal{E}_2 \cap \mathcal{E}_3) = \mathbb{P}(\mathcal{E}_3 \mid \mathcal{E}_2)\mathbb{P}(\mathcal{E}_2) \ge \left( 1 - \frac{d}{e^{\lambda \ell_n/(16L)}} \right) \left( 1 - 2\exp(\frac{-\ell_n}{12}) - 2\exp(\frac{-\sqrt{n\ell_n}}{12}) \right)$.

Then under $\mathcal{E}_2 \cap \mathcal{E}_3$, we have $\|V_{a_n}^{-1}\|_2 \le \|(\sum_{s \in I_{a_n}} x_s x_s^\top)^{-1}\|_2 \le 2\frac{1}{\lambda |I_{a_n}|} \le 4\frac{1}{\lambda \ell_n}$. Then for $i > a_n$, we have

$$\xi_i(x_i) \le \sqrt{L\|V_{a_n}^{-1}\|_2}(\sigma\sqrt{d\log(n + n^2 L/d\beta)} + \sqrt{S\beta})(= g)$$
$$\le \sqrt{L\frac{4}{\lambda \ell_n}}(\sigma\sqrt{d\log(n + n^2 L/d\beta)} + \sqrt{S\beta}). \qquad (11)$$

Here we define $h := \sqrt{L\frac{4}{\lambda \ell_n}}(\sigma\sqrt{d\log(n + n^2 L/d\beta)} + \sqrt{S\beta})$ and $\mathcal{E} := \mathcal{E}_1 \cap \mathcal{E}_2 \cap \mathcal{E}_3$.

We define the set of decision stages until $i$ as $\mathcal{J}_i := [i] \backslash \mathcal{I}_i$ so that $\mathcal{J}_i \cup \mathcal{I}_i = [i]$ and $\mathcal{J}_1 \subseteq \mathcal{J}_2, \ldots, \subseteq \mathcal{J}_n$. Then, we analyze the stopping probability at $i$ in the following lemma.

**Lemma A.8.** *For $i \in \mathcal{J}_n$ with any given $\mathcal{J}_i = \{j_1, j_2, \ldots, j_{|\mathcal{J}_i|}\}$, we have*

$$\mathbb{P}(\tau = i \mid \mathcal{J}_i, \mathcal{E}) = \left( 1 - \frac{1}{n} \right)^{|\mathcal{J}_i| - 1} \frac{1}{n}.$$

*Proof.* For notation simplicity, we define $\mathcal{J}_i^{(k)} := \{j_1, \ldots, j_k\} \subseteq \mathcal{J}_i$ for $k \in [|\mathcal{J}_i|]$. Then, for $i \in \mathcal{J}_n$, we have

$\mathbb{P}(\tau = i \mid \mathcal{J}_i, \mathcal{E})$

$= \mathbb{E}[\mathbb{P}(\tau = i \mid \mathcal{H}_i, \mathcal{J}_i, \mathcal{E}) \mid \mathcal{J}_i, \mathcal{E}]$

$= \mathbb{E}[\mathbb{P}(\{X_t^{\mathrm{LCB}} \leq \alpha \,\forall t \in \mathcal{J}_i^{(|\mathcal{J}_i|-1)}\}, X_i^{\mathrm{LCB}} > \alpha \mid \mathcal{H}_i, \mathcal{J}_i, \mathcal{E}) \mid \mathcal{J}_i, \mathcal{E}]$

$= \mathbb{E}\left[\mathbb{P}(\{X_t^{\mathrm{LCB}} \leq \alpha \,\forall t \in \mathcal{J}_i^{(|\mathcal{J}_i|-1)}\} \mid \mathcal{H}_i, \mathcal{J}_i, \mathcal{E})\mathbb{P}(X_i^{\mathrm{LCB}} > \alpha \mid \mathcal{H}_i, \mathcal{J}_i, \mathcal{E}) \mid \mathcal{J}_i, \mathcal{E}\right]$

$= \mathbb{E}\left[\mathbb{P}(\{X_t^{\mathrm{LCB}} \leq \alpha \,\forall t \in \mathcal{J}_i^{(|\mathcal{J}_i|-1)}\} \mid \mathcal{H}_i, \mathcal{J}_i, \mathcal{E}) \mid \mathcal{J}_i, \mathcal{E}\right]\frac{1}{n}$

$= \mathbb{P}(\{X_t^{\mathrm{LCB}} \leq \alpha \,\forall t \in \mathcal{J}_i^{(|\mathcal{J}_i|-1)}\} \mid \mathcal{J}_i, \mathcal{E})\frac{1}{n}$

$= \mathbb{E}[\mathbb{P}(\{X_t^{\mathrm{LCB}} \leq \alpha \,\forall t \in \mathcal{J}_i^{(|\mathcal{J}_i|-1)}\} \mid \mathcal{H}_{j_{|\mathcal{J}_i|-1}}, \mathcal{J}_i, \mathcal{E}) \mid \mathcal{J}_i, \mathcal{E}]\frac{1}{n}$

$= \mathbb{E}\left[\mathbb{P}(\{X_t^{\mathrm{LCB}} \leq \alpha \,\forall t \in \mathcal{J}_i^{(|\mathcal{J}_i|-2)}\} \mid \mathcal{H}_{j_{|\mathcal{J}_i|-1}}, \mathcal{J}_i, \mathcal{E}) \times \mathbb{P}(X_{j_{|\mathcal{J}_i|-1}}^{\mathrm{LCB}} \leq \alpha \mid \mathcal{H}_{j_{|\mathcal{J}_i|-1}}, \mathcal{J}_i, \mathcal{E}) \mid \mathcal{J}_i, \mathcal{E}\right]\frac{1}{n}$

$= \mathbb{E}\left[\mathbb{P}(\{X_t^{\mathrm{LCB}} \leq \alpha \,\forall t \in \mathcal{J}_i^{(|\mathcal{J}_i|-2)}\} \mid \mathcal{H}_{j_{|\mathcal{J}_i|-1}}, \mathcal{J}_i, \mathcal{E}) \mid \mathcal{J}_i, \mathcal{E}\right]\left(1 - \frac{1}{n}\right)\frac{1}{n}$

$= \mathbb{P}(\{X_t^{\mathrm{LCB}} \leq \alpha \,\forall t \in \mathcal{J}_i^{(|\mathcal{J}_i|-2)}\} \mid \mathcal{J}_i, \mathcal{E})\left(1 - \frac{1}{n}\right)\frac{1}{n}$

$\vdots$

$= \left(1 - \frac{1}{n}\right)^{|\mathcal{J}_i|-1}\frac{1}{n}$

$\square$

From the decision strategy of the algorithm, we have $\mathbb{P}(\tau = i \mid \mathcal{J}_i, \mathcal{E}) = 0$ for all $i \in \mathcal{I}_n$. Therefore, for analyzing $X_\tau$, we have

$\mathbb{E}[X_\tau \mid \mathcal{E}]$

$= \mathbb{E}\left[\sum_{i=1}^{n} \mathbb{P}(\tau = i \mid \mathcal{J}_i, \mathcal{E})\mathbb{E}[X_i \mid \tau = i, \mathcal{J}_i, \mathcal{E}] \mid \mathcal{E}\right]$

$= \mathbb{E}\left[\sum_{i \in \mathcal{J}_n} \mathbb{P}(\tau = i \mid \mathcal{J}_i, \mathcal{E})\mathbb{E}[X_i \mid \tau = i, \mathcal{J}_i, \mathcal{E}] \mid \mathcal{E}\right]$

$\geq \mathbb{E}\left[\sum_{i \in \mathcal{J}_n \setminus [a_n]} \mathbb{P}(\tau = i \mid \mathcal{J}_i, \mathcal{E})\mathbb{E}[X_i \mid \tau = i, \mathcal{J}_i, \mathcal{E}] \mid \mathcal{E}\right]$

$\geq \mathbb{E}\left[\sum_{i \in \mathcal{J}_n \setminus [a_n]} \mathbb{P}(\tau = i \mid \mathcal{J}_i, \mathcal{E})\mathbb{E}\left[\mathbb{E}[X_i^{\mathrm{LCB}} \mid \tau = i, \mathcal{H}_i, \mathcal{J}_i, \mathcal{E}] \mid \tau = i, \mathcal{J}_i, \mathcal{E}\right] \mid \mathcal{E}\right]$

$\geq \mathbb{E}\left[\sum_{i \in \mathcal{J}_n \setminus [a_n]} \left(1 - \frac{1}{n}\right)^{|\mathcal{J}_i|-1}\frac{1}{n}\mathbb{E}\left[\mathbb{E}[X_i^{\mathrm{LCB}} \mid \tau = i, \mathcal{H}_i, \mathcal{J}_i, \mathcal{E}] \mid \tau = i, \mathcal{J}_i, \mathcal{E}\right] \mid \mathcal{E}\right]$

$\geq \mathbb{E}\left[\sum_{i \in \mathcal{J}_n \setminus [a_n]} \left(1 - \frac{1}{n}\right)^{|\mathcal{J}_i|-1}\frac{1}{n}\mathbb{E}\left[\mathbb{E}[X_i^{\mathrm{LCB}} \mid \tau = i, \mathcal{H}_i, \mathcal{J}_i, \mathcal{E}] \mid \tau = i, \mathcal{J}_i, \mathcal{E}\right] \mid \mathcal{E}\right].$ (12)

For the last term above, for $i \in \mathcal{J}_n \setminus [a_n]$, we have

$$\mathbb{E}[X_i^{\text{LCB}} \mid \tau = i, \mathcal{H}_i, \mathcal{J}_i, \mathcal{E}]$$
$$= \mathbb{E}\left[\alpha_i \mid \tau = i, \mathcal{H}_i, \mathcal{J}_i, \mathcal{E}\right] + \mathbb{E}\left[X_i^{\text{LCB}} - \alpha_i \mid X_i^{\text{LCB}} \geq \alpha_i, \mathcal{H}_i, \mathcal{J}_i, \mathcal{E}\right]$$
$$= \mathbb{E}\left[\alpha_i \mid \mathcal{H}_i, \mathcal{E}\right] + \frac{\mathbb{E}[(X_i^{\text{LCB}} - \alpha_i)^+ \mid \mathcal{H}_i, \mathcal{J}_i, \mathcal{E}]}{\mathbb{P}(X_i^{\text{LCB}} \geq \alpha_i \mid \mathcal{H}_i, \mathcal{J}_i, \mathcal{E})}$$
$$\geq \mathbb{E}\left[\alpha^* - 2g \mid \mathcal{H}_i, \mathcal{E}\right] + n\mathbb{E}\left[(X_i - 2g - \alpha^*)^+ \mid \mathcal{H}_i, \mathcal{E}\right] \tag{13}$$

where the last term is obtained from Lemma A.6, $\xi_i(x_i) \leq g$, and the definition of $\alpha_i$.

In what follows, we consider the case of $\mathbb{E}[\max_{i \in [n]} X_i] - \mathcal{O}\left(\sqrt{dL(\sigma^2 d + S)\frac{\log(Ln)}{\lambda \ell_n}}\right) > 0$, because otherwise, it holds trivially:

$$\mathbb{E}[X_\tau \mid \mathcal{E}] \geq \left(\left(1 - \frac{1}{n}\right)^{\sqrt{n\ell_n}} - \left(1 - \frac{1}{n}\right)^{n - \frac{3}{2}\sqrt{n\ell_n} - 1}\right)\left(\mathbb{E}[\max_{i \in [n]} X_i] - \mathcal{O}\left(\sqrt{Ld(\sigma^2 d + S)\frac{\log(Ln)}{\lambda \ell_n}}\right)\right).$$

Recall $h := \sqrt{L\frac{4}{\lambda \ell_n}}(\sigma\sqrt{d\log(n + n^2 L/d\beta)} + \sqrt{S\beta})$. Let i.i.d. $Z_k \sim \mathcal{D}$ for $k \in [n]$. Then, for the last term above in (12) with (13), under $\mathcal{E}$, we have

$$\sum_{i \in \mathcal{J}_n \setminus [a_n]} \left(1 - \frac{1}{n}\right)^{|\mathcal{J}_i| - 1}\frac{1}{n}\left(\mathbb{E}\left[\alpha^* - 2g \mid \mathcal{H}_i, \mathcal{E}\right] + n\mathbb{E}\left[(X_i - 2g - \alpha^*)^+ \mid \mathcal{H}_i, \mathcal{E}\right]\right)$$

$$= \sum_{i \in \mathcal{J}_n \setminus [a_n]} \left(1 - \frac{1}{n}\right)^{|\mathcal{J}_i| - 1}\frac{1}{n}\left(\mathbb{E}\left[\alpha^* - 2g \mid \mathcal{H}_i, \mathcal{E}\right] + n\mathbb{E}\left[(Z_1 - 2g - \alpha^*)^+ \mid \mathcal{H}_i, \mathcal{E}\right]\right)$$

$$\geq \sum_{i \in \mathcal{J}_n \setminus [a_n]} \left(1 - \frac{1}{n}\right)^{|\mathcal{J}_i| - 1}\frac{1}{n}\left(\mathbb{E}\left[\alpha^* - 2g \mid \mathcal{H}_i, \mathcal{E}\right] + \mathbb{E}\left[\max_{k \in [n]}(Z_k - 2g - \alpha^*)^+ \mid \mathcal{H}_i, \mathcal{E}\right]\right)$$

$$\geq \sum_{i \in \mathcal{J}_n \setminus [a_n]} \left(1 - \frac{1}{n}\right)^{|\mathcal{J}_i| - 1}\frac{1}{n}\left(\mathbb{E}\left[\max_{k \in [n]} Z_k \mid \mathcal{H}_i, \mathcal{E}\right] - 4h\right)$$

$$\geq \left(\left(1 - \frac{1}{n}\right)^{\sqrt{n\ell_n}} - \left(1 - \frac{1}{n}\right)^{|\mathcal{J}_n| - 1}\right)\left(\mathbb{E}\left[\max_{k \in [n]} Z_k \mid \mathcal{H}_i, \mathcal{E}\right] - 4h\right)$$

$$= \left(\left(1 - \frac{1}{n}\right)^{\sqrt{n\ell_n}} - \left(1 - \frac{1}{n}\right)^{n - |\mathcal{I}_n| - 1}\right)\left(\mathbb{E}\left[\max_{k \in [n]} Z_k\right] - 4h\right)$$

$$\geq \left(\left(1 - \frac{1}{n}\right)^{\sqrt{n\ell_n}} - \left(1 - \frac{1}{n}\right)^{n - \frac{3}{2}\sqrt{n\ell_n} - 1}\right)\left(\mathbb{E}\left[\max_{k \in [n]} Z_k\right] - 4h\right), \tag{14}$$

where the last inequality is obtained from $\mathcal{E}$

Finally, from (12), (13), and (14), we have

$$\liminf_{n\to\infty} \frac{\mathbb{E}[X_\tau]}{\mathbb{E}[\max_{i\in[n]} X_i]}$$

$$\geq \liminf_{n\to\infty} \frac{\mathbb{E}[X_\tau \mid \mathcal{E}]\mathbb{P}(\mathcal{E})}{\mathbb{E}[\max_{i\in[n]} X_i]}$$

$$\geq \liminf_{n\to\infty} \frac{1}{\mathbb{E}[\max_{i\in[n]} X_i]} \left( \left(1 - \frac{1}{n}\right)^{\sqrt{n\ell_n}} - \left(1 - \frac{1}{n}\right)^{n - \frac{3}{2}\sqrt{n\ell_n} - 1} \right) \left( \mathbb{E}\left[\max_{k\in[n]} X_k\right] - 4h \right) \mathbb{P}(\mathcal{E})$$

$$\geq \liminf_{n\to\infty} \left( \left(1 - \frac{1}{n}\right)^{\sqrt{n\ell_n}} - \left(1 - \frac{1}{n}\right)^{n - \frac{3}{2}\sqrt{n\ell_n} - 1} \right) \left( 1 - \mathcal{O}\left( \frac{1}{\mathbb{E}[\max_{i\in[n]} X_i]} \sqrt{\frac{Ld(\sigma^2 d + S)\log(Ln)}{\lambda\ell_n}} \right) \right) \mathbb{P}(\mathcal{E})$$

$$= \left(1 - \frac{1}{e}\right) \left( 1 - \mathcal{O}\left( \limsup_{n\to\infty} \frac{1}{\mathbb{E}[\max_{i\in[n]} X_i]} \sqrt{\frac{Ld(\sigma^2 d + S)\log(Ln)}{\lambda\ell_n}} \right) \right),$$

where the last equality is obtained from $\ell_n = \Omega(\max\{\frac{L\log d\log n}{\lambda}, \log n\})$ and $\ell_n = o(n)$, and $\mathbb{P}(\mathcal{E}) \geq \left(1 - \frac{1}{n} - \left(1 - \left(1 - \frac{d}{e^{\lambda\ell_n/(16L)}}\right)\left(1 - 2\exp(\frac{-\ell_n}{12}) - 2\exp(\frac{-\sqrt{n\ell_n}}{12})\right)\right)\right)$.

### A.4 PROOF OF PROPOSITION 5.2

**We first provide a proof for** $\frac{\mathbb{E}[x_\tau^\top \theta]}{\mathbb{E}[\max_{i\in[n]} x_i^\top \theta]} \leq \frac{1}{d}$. Let $\theta = (\theta_1, \ldots, \theta_d) \in \mathbb{R}^d$. Consider a non-identical distribution $\mathcal{D}_{x,i}$ that generates the following deterministic points:

$$x_1 = (1, 0, \ldots, 0), \quad x_2 = (0, 1, 0, \ldots, 0), \quad \ldots, \quad x_d = (0, \ldots, 0, 1), \quad x_i = (0, \ldots, 0) \text{ for } i \in \{d+1, \ldots, n\}.$$

For any algorithm, let $\tau$ denote its stopping time.

**Case 1.** Set $\theta_1 = \epsilon$ for $0 < \epsilon < 1$. If $\mathbb{P}(\tau = 1) \leq 1/d$, we set $\theta_2 = \cdots = \theta_d = 0$. Then

$$\mathbb{E}\left[\max_{i\in[n]} x_i^\top \theta\right] = \theta_1, \qquad \mathbb{E}[x_\tau^\top \theta] \leq \frac{\theta_1}{d},$$

so the competitive ratio satisfies $\frac{\mathbb{E}[x_\tau^\top \theta]}{\mathbb{E}[\max_{i\in[n]} x_i^\top \theta]} \leq 1/d$.

**Case 2.** Otherwise if $\mathbb{P}(\tau = 1) > 1/d$ we set $\theta_2 = \theta_1/\epsilon = 1$ for $0 < \epsilon < 1$. If $\mathbb{P}(\tau = 2) \leq 1/d$, then we set $\theta_3 = \theta_4 = \cdots = \theta_d = 0$. Then

$$\mathbb{E}\left[\max_{i\in[n]} x_i^\top \theta\right] = 1, \qquad \mathbb{E}[x_\tau^\top \theta] \leq \frac{\theta_2}{d} + \epsilon = \frac{1}{d} + \epsilon,$$

again yielding, as $\epsilon \to 0$, $\frac{\mathbb{E}[x_\tau^\top \theta]}{\mathbb{E}[\max_{i\in[n]} x_i^\top \theta]} \leq 1/d$.

**Case 3.** Likewise, otherwise if $\mathbb{P}(\tau = 2) > 1/d$, we set $\theta_3 = \theta_2/\epsilon = 1/\epsilon$. If $\mathbb{P}(\tau = 3) \leq 1/d$, then we set $\theta_4 = \theta_5 = \cdots = \theta_d = 0$. We have

$$\mathbb{E}\left[\max_{i\in[n]} x_i^\top \theta\right] = \theta_3 = \frac{1}{\epsilon}, \qquad \mathbb{E}[x_\tau^\top \theta] \leq \frac{1}{\epsilon d} + 1 + \epsilon,$$

again yielding, as $\epsilon \to 0$, $\frac{\mathbb{E}[x_\tau^\top \theta]}{\mathbb{E}[\max_{i\in[n]} x_i^\top \theta]} \leq 1/d$.

There must exist some $i \in \{1, \ldots, d\}$ such that $\mathbb{P}(\tau = i) \leq 1/d$. Therefore, in a similar way, by choosing $\theta$ to place the largest mass of $\theta_{i-1}/\epsilon$ on that coordinate, as $\epsilon \to 0$, we can show that

$$\frac{\mathbb{E}[x_\tau^\top \theta]}{\mathbb{E}[\max_{i\in[n]} x_i^\top \theta]} \leq \frac{1}{d}.$$

Thus, in all cases, one can construct $\theta$ such that the competitive ratio satisfies $\frac{\mathbb{E}[x_\tau^\top \theta]}{\mathbb{E}[\max_{i\in[n]} x_i^\top \theta]} \leq 1/d$.

We next provide a proof for $\frac{\mathbb{E}[x_\tau^\top \theta]}{\mathbb{E}[\max_{i \in [n]} x_i^\top \theta]} \leq \frac{1}{2}$. We can construct $D_{x,1}$ such that $x_1 = (1, 0, \ldots, 0)$ are drawn deterministically. We also consider $\theta = (1, 0, \ldots, 0)$ such that $X_1 = 1$. We also consider $\mathcal{D}_{x,2}$ such that it generates $x_2 = (1/\epsilon, 0, \ldots, 0)$ with probability $\epsilon$ and otherwise, $x_2 = (0, 0, \ldots, 0)$ with probability $1 - \epsilon$. For $i \geq 3$, we consider $x_i = (0, \ldots, 0)$.

Then for any algorithm $\tau$ which does not know $X_i$ for $i \in [n]$ in advance, we have $\mathbb{E}[x_\tau^\top \theta] \leq 1$. On the other hands, the prophet who knows $X_i$ in advance can stop at $\tau = 1$ with $X_1 = 1$ if $X_2 = 0$ with probability $1 - \epsilon$ or stop at $\tau = 2$ if $X_2 = 1/\epsilon$ with probability $\epsilon$. This implies that $\frac{\mathbb{E}[x_\tau^\top \theta]}{\mathbb{E}[\max_{i \in [n]} x_i^\top \theta]} \leq 1/(2 - \epsilon)$. As $\epsilon \to 0$, we can conclude $\frac{\mathbb{E}[x_\tau^\top \theta]}{\mathbb{E}[\max_{i \in [n]} x_i^\top \theta]} \leq 1/2$.

## A.5 PROOF OF THEOREM 5.1

From Lemma A.1, we can show that

$$\mathbb{P}\left(\left|x^\top(\hat{\theta} - \theta)\right| \leq \sqrt{x^\top V^{-1} x}\left(\sigma\sqrt{d \log(n + n\ell_n L/d\beta)} + \sqrt{S\beta}\right), \forall x \in \mathbb{R}^d\right) \geq 1 - 1/n.$$

We define an event $\mathcal{E}_1 = \{|x^\top(\hat{\theta} - \theta)| \leq \sqrt{x^\top V^{-1} x}(\sigma\sqrt{d \log(n + n\ell_n L/d\beta)} + \sqrt{S\beta}), \forall x \in \mathbb{R}^d\}$, which holds with $\mathbb{P}(\mathcal{E}_1) \geq 1 - \frac{1}{n}$. Then under $\mathcal{E}_1$, we have

$$X_i - \xi(x_i) \leq x_i^\top \hat{\theta} \leq X_i + \xi(x_i). \tag{15}$$

**Lemma A.9.** *For $l \geq 1$, let $z_t \sim \mathcal{D}_{x,t}$ for $t \in [l]$ be independent random vectors (not necessarily i.i.d.) satisfying Assumption 3.2. Then*

$$\Pr\left(\frac{1}{l} \sum_{t=1}^{l} z_t z_t^\top \succeq \lambda' I_d\right) \geq 1 - d \exp\left(-\frac{\lambda' l}{8L}\right).$$

*Proof.* Let $\mu_{\min} = \lambda_{\min}(\mathbb{E}[\sum_{t=1}^{l} z_t z_t^\top])$. By the matrix Chernoff bound (Theorem 5.1.1 in Tropp et al. (2015)) for sums of independent PSD matrices with Assumption 3.2, for any $\delta \in [0, 1]$,

$$\Pr\left[\lambda_{\min}\left(\sum_{t=1}^{l} z_t z_t^\top\right) \leq (1 - \delta)\mu_{\min}\right] \leq d\left(\frac{e^{-\delta}}{(1-\delta)^{1-\delta}}\right)^{\mu_{\min}/L} \leq d \exp\left(-\frac{\delta^2}{2} \cdot \frac{\mu_{\min}}{L}\right).$$

Choosing $\delta = \frac{1}{2}$ yields

$$\Pr\left[\lambda_{\min}\left(\sum_{t=1}^{l} z_t z_t^\top\right) \leq \frac{\mu_{\min}}{2}\right] \leq d \exp\left(-\frac{\mu_{\min}}{8L}\right) \leq d \exp\left(-\frac{l\lambda'}{8L}\right),$$

where the last inequality is obtained from Weyl's eigenvalue inequalities. This implies that, with probability at least $1 - d \exp(-\lambda' l/(8L))$,

$$\sum_{t=1}^{l} z_t z_t^\top \succeq \frac{\mu_{\min}}{2} I_d \succeq \frac{l\lambda'}{2} I_d,$$

which completes the proof.

$\square$

Let $\mathcal{E}_2 = \{\sum_{t=1}^{\ell_n} x_t x_t^\top \succeq \frac{\lambda' \ell_n}{2} I_d\}$, which holds with probability at least $1 - \frac{d}{e^{\lambda' \ell_n/(8L)}}$ from Lemma A.9. Then under $\mathcal{E}_2$, we have $\|V^{-1}\|_2 \leq \|(\sum_{t=1}^{\ell_n} x_t x_t^\top)^{-1}\|_2 \leq 2\frac{1}{\lambda' \ell_n}$. Then for $i > \ell_n$, we have

$$\xi(x_i) \leq \sqrt{L\|V^{-1}\|_2}(\sigma\sqrt{d \log(n + n\ell_n L/d\beta)} + \sqrt{S\beta})$$

$$\leq \sqrt{\frac{2L}{\lambda' \ell_n}}(\sigma\sqrt{d \log(n + n\ell_n L/d\beta)} + \sqrt{S\beta}). \tag{16}$$

Here we define $h := \sqrt{\frac{2L}{\lambda' \ell_n}}(\sigma\sqrt{d\log(n + n\ell_n L/d\beta)} + \sqrt{S}\beta)$. Let $z_i \sim \mathcal{D}_{x,i}$ and $\alpha^* = \frac{1}{2}\mathbb{E}\left[\max_{i \in [\ell_n+1,n]} z_i^\top \theta\right]$. Then, from (15) and (16), we have

$$\alpha^* - \frac{1}{2}h \leq \alpha \leq \alpha^* + \frac{1}{2}h. \tag{17}$$

Let $\mathcal{E} := \mathcal{E}_1 \cap \mathcal{E}_2$ and $\mathcal{H}_{\ell_n} = \{\hat{\theta}, V\}$. Then for $i > \ell_n$, we have

$\mathbb{E}[X_\tau^{\mathrm{LCB}}\mathbb{1}(\mathcal{E}) \mid \tau = i, \mathcal{H}_{\ell_n}]\mathbb{P}(\tau = i \mid \mathcal{H}_{\ell_n})$

$= \mathbb{E}[\alpha\mathbb{1}(\mathcal{E}) \mid \mathcal{H}_{\ell_n}]\mathbb{P}(\tau = i \mid \mathcal{H}_{\ell_n}) + \mathbb{E}[X_i^{\mathrm{LCB}}\mathbb{1}(\mathcal{E}) - \alpha\mathbb{1}(\mathcal{E}) \mid \tau = i, \mathcal{H}_{\ell_n}]\mathbb{P}(\tau = i \mid \mathcal{H}_{\ell_n})$

$= \mathbb{E}[\alpha\mathbb{1}(\mathcal{E}) \mid \mathcal{H}_{\ell_n}]\mathbb{P}(\tau = i \mid \mathcal{H}_{\ell_n})$

$\quad + \mathbb{E}[(X_i^{\mathrm{LCB}} - \alpha)_+\mathbb{1}(\mathcal{E}) \mid X_i^{\mathrm{LCB}} \geq \alpha, \mathcal{H}_{\ell_n}]\mathbb{P}(X_i^{\mathrm{LCB}} \geq \alpha \mid \mathcal{H}_{\ell_n}) \prod_{j \in [\ell_n+1,i-1]} \mathbb{P}(X_j^{\mathrm{LCB}} < \alpha \mid \mathcal{H}_{\ell_n})$

$\geq \mathbb{E}[\alpha\mathbb{1}(\mathcal{E}) \mid \mathcal{H}_{\ell_n}]\mathbb{P}(\tau = i \mid \mathcal{H}_{\ell_n}) + \mathbb{E}[(X_i^{\mathrm{LCB}} - \alpha)_+\mathbb{1}(\mathcal{E}) \mid \mathcal{H}_{\ell_n}]\mathbb{P}(\tau = n + 1 \mid \mathcal{H}_{\ell_n})$

$\geq \mathbb{E}[\alpha\mathbb{1}(\mathcal{E}) \mid \mathcal{H}_{\ell_n}]\mathbb{P}(\tau = i \mid \mathcal{H}_{\ell_n})$

$\quad + \mathbb{E}\left[(X_i - 2\xi(x_i) - \alpha)_+ \mathbb{1}(\mathcal{E}) \mid \mathcal{H}_{\ell_n}\right]\mathbb{P}(\tau = n + 1 \mid \mathcal{H}_{\ell_n})$

$\geq \mathbb{E}\left[(\alpha^* - \frac{1}{2}h)\mathbb{1}(\mathcal{E}) \mid \mathcal{H}_{\ell_n}\right]\mathbb{P}(\tau = i \mid \mathcal{H}_{\ell_n})$

$\quad + \left(\mathbb{E}\left[\left(X_i - \alpha^* - \frac{5}{2}h\right)_+ \mathbb{1}(\mathcal{E}) \mid \mathcal{H}_{\ell_n}\right]\right)\mathbb{P}(\tau = n + 1 \mid \mathcal{H}_{\ell_n}),$

where the last inequality is obtained from (17) and $\xi(x_i) \leq h$.

Using the above, we have

$\mathbb{E}[X_\tau \mathbb{1}(\mathcal{E}) \mid \mathcal{H}_{\ell_n}]$

$\geq \sum_{i=1}^n \mathbb{E}\left[\mathbb{E}[X_\tau^{\mathrm{LCB}}\mathbb{1}(\mathcal{E}) \mid \tau = i, \mathcal{H}_{\ell_n}] \cdot \mathbb{P}(\tau = i \mid \mathcal{H}_{\ell_n}) \mid \mathcal{H}_{\ell_n}\right]$

$\geq \sum_{i=\ell_n+1}^n \mathbb{E}\left[\mathbb{E}[X_i^{\mathrm{LCB}}\mathbb{1}(\mathcal{E}) \mid \tau = i, \mathcal{H}_{\ell_n}] \cdot \mathbb{P}(\tau = i \mid \mathcal{H}_{\ell_n}) \mid \mathcal{H}_{\ell_n}\right]$

$\geq \sum_{i=\ell_n+1}^n \mathbb{E}\left[\mathbb{E}\left[(\alpha^* - \frac{1}{2}h)\mathbb{1}(\mathcal{E}) \mid \mathcal{H}_{\ell_n}\right]\mathbb{P}(\tau = i \mid \mathcal{H}_{\ell_n})\right.$

$\quad \left. + \left(\mathbb{E}\left[\left(X_i - \alpha^* - \frac{5}{2}h\right)_+ \mathbb{1}(\mathcal{E}) \mid \mathcal{H}_{\ell_n}\right]\right)\mathbb{P}(\tau = n + 1 \mid \mathcal{H}_{\ell_n}) \mid \mathcal{H}_{\ell_n}\right]$

$\geq \mathbb{E}\left[\left(\mathbb{E}\left[\alpha^*\mathbb{1}(\mathcal{E}) \mid \mathcal{H}_{\ell_n}\right] - \frac{1}{2}h\right)\sum_{i=\ell_n+1}^n \mathbb{P}(\tau = i \mid \mathcal{H}_{\ell_n})\right.$

$\quad \left. + \mathbb{E}\left[\max_{i \in [\ell_n+1,n]}\left(X_i - \alpha^* - \frac{5}{2}h\right)_+ \mathbb{1}(\mathcal{E}) \mid \mathcal{H}_{\ell_n}\right]\mathbb{P}(\tau = n + 1 \mid \mathcal{H}_{\ell_n}) \mid \mathcal{H}_{\ell_n}\right]$

$\geq \mathbb{E}\left[\left(\mathbb{E}\left[\alpha^*\mathbb{1}(\mathcal{E}) \mid \mathcal{H}_{\ell_n}\right] - \frac{1}{2}h\right)(1 - \mathbb{P}(\tau = n + 1 \mid \mathcal{H}_{\ell_n}))\right.$

$\quad \left. + \left(\mathbb{E}\left[\max_{i \in [\ell_n+1,n]} X_i\mathbb{1}(\mathcal{E}) \mid \mathcal{H}_{\ell_n}\right] - \mathbb{E}[\alpha^*\mathbb{1}(\mathcal{E}) \mid \mathcal{H}_{\ell_n}] - \frac{5}{2}h\right)\mathbb{P}(\tau = n + 1 \mid \mathcal{H}_{\ell_n}) \mid \mathcal{H}_{\ell_n}\right]$

$\geq \mathbb{E}\left[(\alpha^*\mathbb{P}(\mathcal{E} \mid \mathcal{H}_{\ell_n}) - \frac{1}{2}h)(1 - \mathbb{P}(\tau = n + 1 \mid \mathcal{H}_{\ell_n}))\right.$

$\quad \left. + \left(\mathbb{E}\left[\max_{i \in [\ell_n+1,n]} X_i\right]\mathbb{P}(\mathcal{E} \mid \mathcal{H}_{\ell_n}) - \alpha^*\mathbb{P}(\mathcal{E} \mid \mathcal{H}_{\ell_n}) - \frac{5}{2}h\right)\mathbb{P}(\tau = n + 1 \mid \mathcal{H}_{\ell_n}) \mid \mathcal{H}_{\ell_n}\right]$

$\geq \alpha^*\mathbb{P}(\mathcal{E} \mid \mathcal{H}_{\ell_n}) - \frac{5}{2}h, \tag{18}$

where the last inequality is obtained from the definition of $\alpha^*$.

Finally, using the above, we have

$$
\begin{aligned}
\liminf_{n\to\infty} \frac{\mathbb{E}[X_\tau]}{\mathbb{E}[\max_{i\in[\ell_n+1,n]} X_i]} &\geq \liminf_{n\to\infty} \frac{\mathbb{E}[\mathbb{E}[X_\tau \mathbb{1}(\mathcal{E})|\mathcal{H}_{\ell_n}]]}{\mathbb{E}[\max_{i\in[\ell_n+1,n]} X_i]} \\
&\geq \liminf_{n\to\infty} \frac{1}{\mathbb{E}[\max_{i\in[\ell_n+1,n]} X_i]} \left(\alpha^*\mathbb{P}(\mathcal{E}) - \frac{5}{2}h\right) \\
&\geq \liminf_{n\to\infty} \left(\frac{1}{2}\left(1 - \frac{1}{n} - \frac{d}{e^{\lambda'\ell_n/8L}}\right) - \mathcal{O}\left(\frac{1}{\mathbb{E}[\max_{i\in[\ell_n+1,n]} X_i]}\sqrt{\frac{L\log(Ln)}{\lambda'\ell_n}}(\sigma\sqrt{d}+\sqrt{S})\right)\right) \\
&= \frac{1}{2} - \mathcal{O}\left(\limsup_{n\to\infty} \frac{1}{\mathbb{E}[\max_{i\in[\ell_n+1,n]} X_i]}\sqrt{\frac{L(\sigma^2 d + S)\log(Ln)}{\lambda'\ell_n}}\right).
\end{aligned}
$$

### A.6 PROOF OF PROPOSITION 5.3

The argument follows the statement used in Marshall et al. (2020). For completeness, we provide the details here. Consider the instance where $X_1 = 1$ deterministically, $X_2 = X_3 = \cdots = X_{n-1} = 0$ deterministically, and $X_n$ takes value $1/\epsilon$ with probability $\epsilon$ (for any $0 < \epsilon < 1$) and 0 otherwise. For any $w_n \leq n-1$, the gambler receives an expected payoff of 1, while the prophet receives an expected payoff of $2 - \epsilon$. Thus, the ratio satisfies $\mathbb{E}[X_\tau]/\mathbb{E}[\max_{i\in[n]} X_i] \leq 1/(2-\epsilon)$. We can conclude the proof with $\epsilon \to 0$.

### A.7 DETAILS OF AN ALGORITHM FOR NON-IID DISTRIBUTIONS UNDER WINDOW ACCESS

In the initial exploration phase of length $\ell_n$, we collect data and estimate $\hat{\theta} = V^{-1}\sum_{t=1}^{\ell_n} y_t x_t$, where $V = \sum_{t=1}^{\ell_n} x_t x_t^\top + \beta I$ for a constant $\beta > 0$. In the subsequent decision phase, we apply LCB-based thresholding with window access for non-identical distributions, as described below.

**Decision with LCB Threshold under Window Access.** Let $z_k \sim \mathcal{D}_{x,k}$ for $k \in [\ell_n + 1, n]$. Then the threshold value is set to

$$
\alpha = \tfrac{1}{2}\mathbb{E}\left[\max\left\{\max_{i\in[\ell_n]} x_i^\top\hat{\theta}, \max_{k\in[\ell_n+1,n]} z_k^\top\hat{\theta}\right\}\Big|\mathcal{F}_{\ell_n}\right], \tag{19}
$$

and we define LCBs as

$$
X_i^{\mathrm{LCB}} = x_i^\top\hat{\theta} - \xi_i(x_i), \tag{20}
$$

where $\xi_i(x_i) := \sqrt{x_i^\top V^{-1} x_i}\left(\sigma\sqrt{d\log(n + n\ell_n L/d\beta)} + \sqrt{S\beta}\right)$.

At stage $\ell_n + 1$, the algorithm checks whether $\max_{k\in[1,\ell_n+1]} X_k^{\mathrm{LCB}} \geq \alpha$. If so, it stops with $\tau = \arg\max_{k\in[1,\ell_n+1]} X_k^{\mathrm{LCB}}$; otherwise, it continues. For $i > \ell_n + 1$, the algorithm stops at stage $i$ if $X_i^{\mathrm{LCB}} \geq \alpha$.

### A.8 PROOF OF THEOREM 5.4

From Lemma A.1, we can show that

$$
\mathbb{P}\left(\left|x^\top(\hat{\theta}-\theta)\right| \leq \sqrt{x^\top V^{-1} x}\left(\sigma\sqrt{d\log(n + n\ell_n L/d\beta)} + \sqrt{S\beta}\right), \forall x \in \mathbb{R}^d\right) \geq 1 - 1/n.
$$

We define an event $\mathcal{E}_1 = \{|x^\top(\hat{\theta}-\theta)| \leq \sqrt{x^\top V^{-1} x}(\sigma\sqrt{d\log(n + n\ell_n L/d\beta)} + \sqrt{S\beta}), \forall x \in \mathbb{R}^d\}$, which holds with $\mathbb{P}(\mathcal{E}_1) \geq 1 - \frac{1}{n}$.

Let $\mathcal{E}_2 = \{\sum_{t\in[\ell_n]} x_t x_t^\top \succeq \frac{\lambda'\ell_n}{2} I_d\}$, which holds with probability at least $1 - \frac{d}{e^{\lambda'\ell_n/8L}}$ from Lemma A.9. Then under $\mathcal{E}_2$, we have $\|V^{-1}\|_2 \leq \|(\sum_{t\in[\ell_n]} x_t x_t^\top)^{-1}\|_2 \leq 2\frac{1}{\lambda'\ell_n}$. Then for all

---

**Algorithm 3** Explore-Then-Decide with LCB Thresholding under Window Access (`ETD-LCBT-WA`)

---

**Input:** Exploration length $\ell_n$; regularization parameter $\beta$
**Output:** Stopping time $\tau$

1 **for** $i = 1, \ldots, n$ **do**
2      **if** $i \leq \ell_n$ **then**
3         Observe $(x_i, y_i)$
4      **else if** $i = \ell_n + 1$ **then**
5         Observe $(x_i, y_i)$
6         Compute $\alpha$ from (19) and $X_k^{\mathrm{LCB}}$ for $k \leq \ell_n + 1$ from (20).
7         **if** $\max_{k \in [1, \ell_n+1]} X_k^{\mathrm{LCB}} \geq \alpha$ **then**
8            Stop with $\tau \leftarrow \arg\max_{k \in [1, \ell_n+1]} X_k^{\mathrm{LCB}}$
9      **else**
10        Observe $(x_i, y_i)$
11        Compute $X_i^{\mathrm{LCB}}$ from (20).
12        **if** $X_i^{\mathrm{LCB}} \geq \alpha$ **then**
13           Stop with $\tau \leftarrow i$

---

$i \in [n]$, we have

$$\xi_i(x_i) \leq \sqrt{L\|V^{-1}\|_2}(\sigma\sqrt{d\log(n + n\ell_n L/d\beta)} + \sqrt{S\beta})$$

$$\leq \sqrt{L\frac{2}{\lambda'\ell_n}}(\sigma\sqrt{d\log(n + n^2 L/d\beta)} + \sqrt{S\beta}). \tag{21}$$

Here we define $h := \sqrt{L\frac{2}{\lambda'\ell_n}}(\sigma\sqrt{d\log(n + n^2 L/d\beta)} + \sqrt{S\beta})$, $\mathcal{E} := \mathcal{E}_1 \cap \mathcal{E}_2$. Then, on $\mathcal{E}$, for $x \sim \mathcal{D}_{x,i}$ for any $i \in [n]$, we have

$$x^\top\theta - h \leq x^\top\theta - \xi(x) \leq x^\top\hat{\theta} \leq x^\top\theta + \xi(x) \leq x^\top\theta + h.$$

Also, for all $i \in [n]$,

$$X_i - 2h \leq X_i^{\mathrm{LCB}} \leq X_i.$$

We aggregate the whole window into a single random variable. Define the window-maximum LCB at time $\ell_n + 1$:

$$\widetilde{X}_1^{\mathrm{LCB}} := \max_{k \in [1, \ell_n+1]} X_k^{\mathrm{LCB}}.$$

For $r = 2, \ldots, m$ where $m := n - \ell_n$, define

$$\widetilde{X}_r^{\mathrm{LCB}} := X_{\ell_n+r}^{\mathrm{LCB}}.$$

Define the associated threshold stopping time on the reduced sequence:

$$\widetilde{\tau} := \inf\{r \in [m] : \widetilde{X}_r^{\mathrm{LCB}} \geq \alpha\}, \qquad \widetilde{\tau} = m + 1 \text{ if the set is empty.}$$

Then Algorithm 3 is equivalent to: (i) stop at $r = 1$ if $\widetilde{X}_1^{\mathrm{LCB}} \geq \alpha$ and return the argmax in $[1, \ell_n+1]$, otherwise (ii) stop at the first $r \geq 2$ such that $\widetilde{X}_r^{\mathrm{LCB}} \geq \alpha$. In particular, the realized LCB payoff satisfies

$$X_\tau^{\mathrm{LCB}} = \widetilde{X}_{\widetilde{\tau}}^{\mathrm{LCB}}, \quad \text{with the convention } \widetilde{X}_{m+1}^{\mathrm{LCB}} := 0. \tag{22}$$

We also define $\widetilde{X}_1 = \max_{k \in [1, \ell_n+1]} X_k$ and $\widetilde{X}_r = X_{\ell_n+r}$ for $r \in [2, m]$. Let $\mathcal{F}_{\ell_n} := \sigma\left((x_t, y_t)_{t=1}^{\ell_n}\right)$ be the full sigma-field generated by the exploration data and $Z_i = z_i^\top\theta$ for $z_i \sim \mathcal{D}_{x,i}$. By the fact that $(Z_i)_{i \geq \ell_n+1}$ has the same conditional law as $(X_i)_{i \geq \ell_n+1}$ given $\mathcal{F}_{\ell_n}$, on $\mathcal{E}$, we have

$$2\alpha = \mathbb{E}\left[\max\left\{\max_{i \in [\ell_n]} x_i^\top\hat{\theta}, \max_{k \in [\ell_n+1, n]} z_k^\top\hat{\theta}\right\} \,\middle|\, \mathcal{F}_{\ell_n}\right] \leq \mathbb{E}\left[\max\left\{\max_{i \in [\ell_n]} X_i, \max_{k \in [\ell_n+1, n]} Z_k\right\} \,\middle|\, \mathcal{F}_{\ell_n}\right] + h$$

$$= \mathbb{E}\left[\max_{r \in [m]} \widetilde{X}_r \,\middle|\, \mathcal{F}_{\ell_n}\right] + h. \tag{23}$$

Then for $r \in [m]$, we have

$$\mathbb{E}[\widetilde{X}_{\widetilde{\tau}}^{\mathrm{LCB}}\mathbb{1}(\mathcal{E}) \mid \widetilde{\tau} = r, \mathcal{F}_{\ell_n}]\mathbb{P}(\widetilde{\tau} = r \mid \mathcal{F}_{\ell_n})$$

$$= \mathbb{E}[\alpha\mathbb{1}(\mathcal{E}) \mid \mathcal{F}_{\ell_n}]\mathbb{P}(\widetilde{\tau} = r \mid \mathcal{F}_{\ell_n}) + \mathbb{E}[\widetilde{X}_r^{\mathrm{LCB}}\mathbb{1}(\mathcal{E}) - \alpha\mathbb{1}(\mathcal{E}) \mid \widetilde{\tau} = r, \mathcal{F}_{\ell_n}]\mathbb{P}(\widetilde{\tau} = r \mid \mathcal{F}_{\ell_n})$$

$$= \mathbb{E}[\alpha\mathbb{1}(\mathcal{E}) \mid \mathcal{F}_{\ell_n}]\mathbb{P}(\widetilde{\tau} = r \mid \mathcal{F}_{\ell_n})$$

$$\quad + \mathbb{E}[(\widetilde{X}_r^{\mathrm{LCB}} - \alpha)_+\mathbb{1}(\mathcal{E}) \mid \widetilde{X}_r^{\mathrm{LCB}} \geq \alpha, \mathcal{F}_{\ell_n}]\mathbb{P}(\widetilde{X}_r^{\mathrm{LCB}} \geq \alpha \mid \mathcal{F}_{\ell_n})\mathbb{P}(\widetilde{X}_j^{\mathrm{LCB}} < \alpha, \forall j \in [r-1] \mid \mathcal{F}_{\ell_n})$$

$$\geq \mathbb{E}[\alpha\mathbb{1}(\mathcal{E}) \mid \mathcal{F}_{\ell_n}]\mathbb{P}(\widetilde{\tau} = r \mid \mathcal{F}_{\ell_n}) + \mathbb{E}[(\widetilde{X}_r^{\mathrm{LCB}} - \alpha)_+\mathbb{1}(\mathcal{E}) \mid \mathcal{F}_{\ell_n}]\mathbb{P}(\widetilde{\tau} = m+1 \mid \mathcal{F}_{\ell_n})$$

$$\geq \mathbb{E}[\alpha\mathbb{1}(\mathcal{E}) \mid \mathcal{F}_{\ell_n}]\mathbb{P}(\widetilde{\tau} = r \mid \mathcal{F}_{\ell_n})$$

$$\quad + \mathbb{E}\left[\left(\widetilde{X}_r - 2h - \alpha\right)_+ \mathbb{1}(\mathcal{E}) \mid \mathcal{F}_{\ell_n}\right]\mathbb{P}(\widetilde{\tau} = m+1 \mid \mathcal{F}_{\ell_n}). \tag{24}$$

Using the above, summing over $r \in [m]$,

$$\mathbb{E}\left[\widetilde{X}_{\widetilde{\tau}}^{\mathrm{LCB}}\mathbb{1}(\mathcal{E}) \mid \mathcal{F}_{\ell_n}\right]$$

$$\geq \alpha\mathbb{1}(\mathcal{E})\mathbb{P}(\widetilde{\tau} \leq m \mid \mathcal{F}_{\ell_n}) + \mathbb{1}(\mathcal{E})\mathbb{P}(\widetilde{\tau} = m+1 \mid \mathcal{F}_{\ell_n})\mathbb{E}\left[\sum_{r \in [m]} (\widetilde{X}_r - 2h - \alpha)_+ \mid \mathcal{F}_{\ell_n}\right]$$

$$\geq \alpha\mathbb{1}(\mathcal{E})\mathbb{P}(\widetilde{\tau} \leq m \mid \mathcal{F}_{\ell_n}) + \mathbb{1}(\mathcal{E})\mathbb{P}(\widetilde{\tau} = m+1 \mid \mathcal{F}_{\ell_n})\mathbb{E}\left[\max_{r \in [m]}(\widetilde{X}_r - 2h - \alpha)_+ \mid \mathcal{F}_{\ell_n}\right]. \tag{25}$$

Using $\max_r(\widetilde{X}_r - 2h - \alpha)_+ \geq \max_r \widetilde{X}_r - 2h - \alpha$ and (23), the RHS of (25) becomes

$$\geq \mathbb{1}(\mathcal{E})\left(\alpha(1-p) + p\left(\mathbb{E}[\max_r \widetilde{X}_r \mid \mathcal{F}_{\ell_n}] - \alpha - 2h\right)\right)$$

$$\geq \mathbb{1}(\mathcal{E})\left(\alpha(1-p) + p(2\alpha - \alpha - 3h)\right)$$

$$\geq \mathbb{1}(\mathcal{E})\alpha - 3h,$$

where $p := \mathbb{P}(\widetilde{\tau} = m+1 \mid \mathcal{F}_{\ell_n})$. Hence, taking expectation gives

$$\mathbb{E}\left[\widetilde{X}_{\widetilde{\tau}}^{\mathrm{LCB}}\mathbb{1}(\mathcal{E})\right] \geq \mathbb{E}[\alpha\,\mathbb{1}(\mathcal{E})] - 3h.$$

On $\mathcal{E}$, we have $X_\tau \geq X_\tau^{\mathrm{LCB}} = \widetilde{X}_{\widetilde{\tau}}^{\mathrm{LCB}}$. Therefore,

$$\mathbb{E}[X_\tau] \geq \mathbb{E}[X_\tau\mathbb{1}(\mathcal{E})] \geq \mathbb{E}[\widetilde{X}_{\widetilde{\tau}}^{\mathrm{LCB}}\mathbb{1}(\mathcal{E})] \geq \mathbb{E}[\alpha\,\mathbb{1}(\mathcal{E})] - 3h. \tag{26}$$

Also we have

$$\mathbb{E}[\alpha\,\mathbb{1}(\mathcal{E})] \geq \frac{1}{2}\mathbb{E}\left[\max_{r \in [m]} \widetilde{X}_r\,\mathbb{1}(\mathcal{E})\right] - \frac{1}{2}h = \frac{1}{2}\mathbb{E}\left[\max_{i \in [n]} X_i\,\mathbb{1}(\mathcal{E})\right] - \frac{1}{2}h.$$

Let $M := \max_{i \in [n]} X_i$. Since $0 \leq M \leq \sqrt{LS}$, we have

$$\mathbb{E}[M\mathbb{1}(\mathcal{E})] \geq \mathbb{E}[M] - \sqrt{LS}\,\mathbb{P}(\mathcal{E}^c).$$

Thus,

$$\mathbb{E}[\alpha\,\mathbb{1}(\mathcal{E})] \geq \frac{1}{2}\mathbb{E}[M\,\mathbb{1}(\mathcal{E})] - \frac{1}{2}h \geq \frac{1}{2}\left(\mathbb{E}[M] - \sqrt{LS}\,\mathbb{P}(\mathcal{E}^c) - h\right).$$

Combining with (26) yields

$$\mathbb{E}[X_\tau] \geq \frac{1}{2}\mathbb{E}\left[\max_{i \in [n]} X_i\right] - O(h) - \frac{\sqrt{LS}}{2}\mathbb{P}(\mathcal{E}^c). \tag{27}$$

Finally, since $\mathbb{P}(\mathcal{E}^c) \leq \frac{1}{n} + d\exp(-\lambda'\ell_n/(8L))$, we get

$$\frac{\mathbb{E}[X_\tau]}{\mathbb{E}[\max_i X_i]} \geq \frac{1}{2} - O\left(\frac{1}{\mathbb{E}[\max_i X_i]}\left(\sqrt{\frac{L(\sigma^2 d + S)\log(Ln)}{\lambda'\ell_n}} + \sqrt{LS}\left(\frac{1}{n} + de^{-\lambda'\ell_n/(8L)}\right)\right)\right).$$

---

**Algorithm 4** Decide with Offline Samples and LCB Thresholding (`DOS-LCBT`)

**Input:** Offline samples $(x_t^o, y_t^o)$ for $t \in \mathcal{S}$; regularization parameter $\beta$
**Output:** Stopping time $\tau$

1   $V \leftarrow \sum_{t \in \mathcal{S}} x_t^o x_t^{o\top} + \beta I_d; \ \hat{\theta} \leftarrow V^{-1} \sum_{t \in \mathcal{S}} y_t^o x_t^o$
2   Compute $\alpha$ from (28)
3   **for** $i = 1, \ldots, n$ **do**
4      Observe $(y_i, x_i)$
5      Compute $X_i^{\text{LCB}}$ from (1)
6      **if** $X_i^{\text{LCB}} \geq \alpha$ **then**
7         $\lfloor$ Stop and set $\tau \leftarrow i$

---

### A.9   FURTHER DETAILS ON THE METHOD USING OFFLINE SAMPLES UNDER NON-I.I.D. DISTRIBUTIONS

For any given $\mathcal{S} \subseteq [n]$ such that $|S| = \ell_n$ for $\ell_n > 0$ (specified later), we assume that the gambler receives offline samples $(x_t^o, y_t^o)$ where $x_t^o \sim \mathcal{D}_{x,t}$ for $t \in \mathcal{S}$ and $y_t^o = x_t^{o\top}\theta + \eta_t$. In Algorithm 4, using this offline samples, we obtain $V = \sum_{t \in \mathcal{S}} x_t^o x_t^{o\top} + \beta I_d$ and $\hat{\theta} = V^{-1} \sum_{t \in \mathcal{S}} y_t^o x_t^o$ for constant $\beta > 0$. Then we follow the following decision strategy.

**Decision with LCB Threshold.**   For each time $i \geq 1$, for $z_s \sim \mathcal{D}_{x,s}$ for all $s \in [1, n]$, we define the threshold:

$$\alpha = \frac{1}{2}\mathbb{E}\left[\max_{s \in [n]} z_s^\top \hat{\theta} \mid \hat{\theta}\right] \tag{28}$$

Recall the lower confidence bound for $X_i$ in the Explore-then-Decide framework: $X_i^{LCB} = x_i^\top \hat{\theta} - \xi(x_i)$, where $\xi(x_i) := \sqrt{x_i^\top V^{-1} x_i}(\sigma\sqrt{d \log(n + n\ell_n L/d\beta)} + \sqrt{S\beta})$. The algorithm stops at stage $i$ if $X_i^{LCB} \geq \alpha$.

**Theorem A.10.** *In the non-i.i.d. setting with unknown distributions and $\ell_n$ offline samples of $\mathcal{S}$, Algorithm 4 with $\ell_n = o(n)$, $\ell_n = \omega(\frac{L \log d}{\lambda'})$, and a constant $\beta > 0$ achieves the following asymptotic competitive ratio:*

$$\liminf_{n \to \infty} \frac{\mathbb{E}[X_\tau]}{\mathbb{E}[\max_{i \in [n]} X_i]} \geq \frac{1}{2} - \mathcal{O}\left(\limsup_{n \to \infty} \frac{1}{\mathbb{E}[\max_{i \in [n]} X_i]}\sqrt{\frac{L(\sigma^2 d + S)\log(Ln)}{\lambda' \ell_n}}\right).$$

*Furthermore, by setting $\ell_n = \frac{L(\sigma^2 + S)}{\lambda} f(n) \log(Ln)$ for some function $f(n)$ (e.g., $f(n) = \Theta(\log^p n)$ for $p > 0$, or $\Theta(n^q)$ for $0 < q < 1$) satisfying $\ell_n = o(n)$, if $\mathrm{OPT} = \omega(1/\sqrt{f(n)})$, then Algorithm 4 achieves the following asymptotic competitive ratio:*

$$\liminf_{n \to \infty} \frac{\mathbb{E}[X_\tau]}{\mathbb{E}[\max_{i \in [n]} X_i]} \geq \frac{1}{2}.$$

*Proof.* From Lemma A.1, we can show that

$$\mathbb{P}\left(\left|x^\top(\hat{\theta} - \theta)\right| \leq \sqrt{x^\top V^{-1} x}\left(\sigma\sqrt{d \log(n + n\ell_n L/d\beta)} + \sqrt{S\beta}\right), \forall x \in \mathbb{R}^d\right) \geq 1 - 1/n.$$

We define an event $\mathcal{E}_1 = \{|x^\top(\hat{\theta} - \theta)| \leq \sqrt{x^\top V^{-1} x}(\sigma\sqrt{d \log(n + n\ell_n L/d\beta)} + \sqrt{S\beta}), \forall x \in \mathbb{R}^d\}$, which holds with $\mathbb{P}(\mathcal{E}_1) \geq 1 - \frac{1}{n}$. Then under $\mathcal{E}_1$, we have

$$X_i - \xi(x_i) \leq x_i^\top \hat{\theta} \leq X_i + \xi(x_i). \tag{29}$$

**Lemma A.11.** *For any $\mathcal{S} \subset [n]$ with $|\mathcal{S}| = l$ for $l > 0$, let $z_t \sim \mathcal{D}_{x,t}$ for $t \in \mathcal{S}$ be independent random vectors (not necessarily i.i.d.) satisfying Assumption 3.2. Then*

$$\Pr\left(\frac{1}{l}\sum_{t \in \mathcal{S}} z_t z_t^\top \succeq \lambda' I_d\right) \geq 1 - d\exp\left(-\frac{\lambda' l}{8L}\right).$$

*Proof.* Let $\mu_{\min} = \lambda_{\min}(\mathbb{E}[\sum_{t \in \mathcal{S}} z_t z_t^\top])$. By the matrix Chernoff bound (Theorem 5.1.1 in Tropp et al. (2015)) for sums of independent PSD matrices with Assumption 3.2, for any $\delta \in [0, 1]$,

$$\Pr\left[\lambda_{\min}\left(\sum_{t \in \mathcal{S}} z_t z_t^\top\right) \le (1-\delta)\mu_{\min}\right] \le d\left(\frac{e^{-\delta}}{(1-\delta)^{1-\delta}}\right)^{\mu_{\min}/L} \le d\exp\left(-\frac{\delta^2}{2} \cdot \frac{\mu_{\min}}{L}\right).$$

Choosing $\delta = \frac{1}{2}$ yields

$$\Pr\left[\lambda_{\min}\left(\sum_{t \in \mathcal{S}} z_t z_t^\top\right) \le \frac{\mu_{\min}}{2}\right] \le d\exp\left(-\frac{\mu_{\min}}{8L}\right) \le d\exp\left(-\frac{l\lambda'}{8L}\right),$$

where the last inequality is obtained from Weyl's eigenvalue inequalities. Equivalently, with probability at least $1 - d\exp(-\lambda' l/(8L))$,

$$\sum_{t \in \mathcal{S}} z_t z_t^\top \succeq \frac{\mu_{\min}}{2} I_d \succeq \frac{l\lambda'}{2} I_d,$$

which completes the proof.

$\square$

Let $\mathcal{E}_2 = \{\sum_{t \in \mathcal{S}} x_t^o x_t^{o\top} \succeq \frac{\lambda' \ell_n}{2} I_d\}$, which holds with probability at least $1 - \frac{d}{e^{\lambda' \ell_n/8L}}$ from Lemma A.11. Then under $\mathcal{E}_2$, we have $\|V^{-1}\|_2 \le \|(\sum_{t \in \mathcal{S}} x_t^o x_t^{o\top})^{-1}\|_2 \le 2\frac{1}{\lambda' \ell_n}$. Then for $i \in [n]$, we have

$$\xi(x_i) \le \sqrt{\|x_i\|_2^2 \|V^{-1}\|_2}(\sigma\sqrt{d\log(n + n\ell_n L/d\beta)} + \sqrt{S}\beta)(:= g_i)$$

$$\le \sqrt{\frac{2L}{\lambda' \ell_n}}(\sigma\sqrt{d\log(n + n\ell_n L/d\beta)} + \sqrt{S}\beta). \tag{30}$$

Here we define $h := \sqrt{\frac{2L}{\lambda' \ell_n}}(\sigma\sqrt{d\log(n + n\ell_n L/d\beta)} + \sqrt{S}\beta)$. Let $z_i \sim \mathcal{D}_{x,i}$ and $\alpha^* = \frac{1}{2}\mathbb{E}\left[\max_{i \in [n]} z_i^\top \theta\right]$. Then, from (29) and (30), we have

$$\alpha^* - \frac{1}{2}h \le \alpha \le \alpha^* + \frac{1}{2}h. \tag{31}$$

Let $\mathcal{E} := \mathcal{E}_1 \cap \mathcal{E}_2$ and $\mathcal{H}_{\ell_n} = \{\hat{\theta}, V\}$. Then for $i \ge 1$, we have

$$\mathbb{E}[X_\tau^{\mathrm{LCB}} \mathbb{1}(\mathcal{E}) \mid \tau = i, \mathcal{H}_{\ell_n}]\mathbb{P}(\tau = i \mid \mathcal{H}_{\ell_n})$$
$$= \mathbb{E}[\alpha \mathbb{1}(\mathcal{E}) \mid \mathcal{H}_{\ell_n}]\mathbb{P}(\tau = i \mid \mathcal{H}_{\ell_n}) + \mathbb{E}[X_i^{\mathrm{LCB}} \mathbb{1}(\mathcal{E}) - \alpha \mathbb{1}(\mathcal{E}) \mid \tau = i, \mathcal{H}_{\ell_n}]\mathbb{P}(\tau = i \mid \mathcal{H}_{\ell_n})$$
$$= \mathbb{E}[\alpha \mathbb{1}(\mathcal{E}) \mid \mathcal{H}_{\ell_n}]\mathbb{P}(\tau = i \mid \mathcal{H}_{\ell_n})$$
$$\quad + \mathbb{E}[(X_i^{\mathrm{LCB}} - \alpha)_+ \mathbb{1}(\mathcal{E}) \mid X_i^{\mathrm{LCB}} \ge \alpha, \mathcal{H}_{\ell_n}]\mathbb{P}(X_i^{\mathrm{LCB}} \ge \alpha \mid \mathcal{H}_{\ell_n}) \prod_{j \in [i-1]} \mathbb{P}(X_j^{\mathrm{LCB}} < \alpha_j | \mathcal{H}_{\ell_n})$$
$$\ge \mathbb{E}[\alpha \mathbb{1}(\mathcal{E}) \mid \mathcal{H}_{\ell_n}]\mathbb{P}(\tau = i \mid \mathcal{H}_{\ell_n}) + \mathbb{E}[(X_i^{\mathrm{LCB}} - \alpha)_+ \mathbb{1}(\mathcal{E}) \mid \mathcal{H}_{\ell_n}]\mathbb{P}(\tau = n + 1 \mid \mathcal{H}_{\ell_n})$$
$$\ge \mathbb{E}[\alpha \mathbb{1}(\mathcal{E}) \mid \mathcal{H}_{\ell_n}]\mathbb{P}(\tau = i \mid \mathcal{H}_{\ell_n})$$
$$\quad + \mathbb{E}\left[(X_i - 2\xi(x_i) - \alpha)_+ \mathbb{1}(\mathcal{E}) \mid \mathcal{H}_{\ell_n}\right]\mathbb{P}(\tau = n + 1 \mid \mathcal{H}_{\ell_n})$$
$$\ge \mathbb{E}\left[(\alpha^* - \frac{1}{2}h)\mathbb{1}(\mathcal{E}) \mid \mathcal{H}_{\ell_n}\right]\mathbb{P}(\tau = i \mid \mathcal{H}_{\ell_n})$$
$$\quad + \left(\mathbb{E}\left[\left(X_i - \alpha^* - \frac{5}{2}h\right)_+ \mathbb{1}(\mathcal{E}) \mid \mathcal{H}_{\ell_n}\right]\right)\mathbb{P}(\tau = n + 1 \mid \mathcal{H}_{\ell_n}),$$

where the last inequality is obtained from (31) and $\xi(x_i) \le h$.

Using the above, we have

$$\mathbb{E}[X_\tau \mathbb{1}(\mathcal{E}) \mid \mathcal{H}_{\ell_n}]$$

$$\geq \sum_{i=1}^n \mathbb{E}\left[\mathbb{E}[X_\tau^{\mathrm{LCB}}\mathbb{1}(\mathcal{E}) \mid \tau = i, \mathcal{H}_{\ell_n}] \cdot \mathbb{P}(\tau = i \mid \mathcal{H}_{\ell_n}) \mid \mathcal{H}_{\ell_n}\right]$$

$$= \sum_{i=1}^n \mathbb{E}\left[\mathbb{E}[X_i^{\mathrm{LCB}}\mathbb{1}(\mathcal{E}) \mid \tau = i, \mathcal{H}_{\ell_n}] \cdot \mathbb{P}(\tau = i \mid \mathcal{H}_{\ell_n}) \mid \mathcal{H}_{\ell_n}\right]$$

$$\geq \sum_{i=1}^n \mathbb{E}\left[\mathbb{E}\left[(\alpha^* - \frac{1}{2}h)\mathbb{1}(\mathcal{E}) \mid \mathcal{H}_{\ell_n}\right]\mathbb{P}(\tau = i \mid \mathcal{H}_{\ell_n})\right.$$
$$\left. + \left(\mathbb{E}\left[\left(X_i - \alpha^* - \frac{5}{2}h\right)_+ \mathbb{1}(\mathcal{E}) \mid \mathcal{H}_{\ell_n}\right]\right)\mathbb{P}(\tau = n + 1 \mid \mathcal{H}_{\ell_n}) \mid \mathcal{H}_{\ell_n}\right]$$

$$\geq \mathbb{E}\left[\left(\mathbb{E}\left[\alpha^*\mathbb{1}(\mathcal{E}) \mid \mathcal{H}_{\ell_n}\right] - \frac{1}{2}h\right)\sum_{i=1}^n \mathbb{P}(\tau = i \mid \mathcal{H}_{\ell_n})\right.$$
$$\left. + \mathbb{E}\left[\max_{i\in[n]}\left(X_i - \alpha^* - \frac{5}{2}h\right)_+ \mathbb{1}(\mathcal{E}) \mid \mathcal{H}_{\ell_n}\right]\mathbb{P}(\tau = n + 1 \mid \mathcal{H}_{\ell_n}) \mid \mathcal{H}_{\ell_n}\right]$$

$$\geq \mathbb{E}\left[\left(\mathbb{E}\left[\alpha^*\mathbb{1}(\mathcal{E}) \mid \mathcal{H}_{\ell_n}\right] - \frac{1}{2}h\right)(1 - \mathbb{P}(\tau = n + 1 \mid \mathcal{H}_{\ell_n}))\right.$$
$$\left. + \left(\mathbb{E}\left[\max_{i\in[n]}X_i\mathbb{1}(\mathcal{E}) \mid \mathcal{H}_{\ell_n}\right] - \mathbb{E}[\alpha^*\mathbb{1}(\mathcal{E}) \mid \mathcal{H}_{\ell_n}] - \frac{5}{2}h\right)\mathbb{P}(\tau = n + 1 \mid \mathcal{H}_{\ell_n}) \mid \mathcal{H}_{\ell_n}\right]$$

$$\geq \mathbb{E}\left[(\alpha^*\mathbb{P}(\mathcal{E} \mid \mathcal{H}_{\ell_n}) - \frac{1}{2}h)(1 - \mathbb{P}(\tau = n + 1 \mid \mathcal{H}_{\ell_n}))\right.$$
$$\left. + \left(\mathbb{E}[\max_{i\in[n]}X_i]\mathbb{P}(\mathcal{E} \mid \mathcal{H}_{\ell_n}) - \alpha^*\mathbb{P}(\mathcal{E} \mid \mathcal{H}_{\ell_n}) - \frac{5}{2}h\right)\mathbb{P}(\tau = n + 1 \mid \mathcal{H}_{\ell_n}) \mid \mathcal{H}_{\ell_n}\right]$$

$$\geq \alpha^*\mathbb{P}(\mathcal{E} \mid \mathcal{H}_{\ell_n}) - \frac{5}{2}h.$$

Finally, using the above, we have

$$\liminf_{n\to\infty} \frac{\mathbb{E}[X_\tau]}{\mathbb{E}[\max_{i\in[n]}X_i]} \geq \liminf_{n\to\infty} \frac{\mathbb{E}[\mathbb{E}[X_\tau\mathbb{1}(\mathcal{E})|\mathcal{H}_{\ell_n}]]}{\mathbb{E}[\max_{i\in[n]}X_i]}$$

$$\geq \liminf_{n\to\infty} \frac{1}{\mathbb{E}[\max_{i\in[n]}X_i]}\left(\alpha^*\mathbb{P}(\mathcal{E}) - \frac{5}{2}h\right)$$

$$\geq \liminf_{n\to\infty}\left(\frac{1}{2}\left(1 - \frac{1}{n} - \frac{d}{e^{\lambda'\ell_n/8L}}\right) - \mathcal{O}\left(\frac{1}{\mathbb{E}[\max_{i\in[n]}X_i]}\sqrt{\frac{L\log(Ln)}{\lambda'\ell_n}}(\sigma\sqrt{d} + \sqrt{S})\right)\right)$$

$$= \frac{1}{2} - \mathcal{O}\left(\limsup_{n\to\infty}\frac{1}{\mathbb{E}[\max_{i\in[n]}X_i]}\sqrt{\frac{L(\sigma^2 d + S)\log(Ln)}{\lambda'\ell_n}}\right).$$

$$\square$$

