# OpenReview forum: "Learning in Prophet Inequalities with Noisy Observations"
_ICLR.cc/2026/Conference — ICLR 2026 Poster_

### Official Review · Reviewer_YYur · 2025-10-18

**Soundness:** 3
**Presentation:** 3
**Contribution:** 3
**Rating:** 4
**Confidence:** 2

**Summary:**

The paper studies Prophet Inequalities under a novel setting. In such a setting, the rewards at time $i\in[n]$ are defined as $x_i^\top \theta$ where $\theta$ is an unknown fixed latent vector, while $x_i$ is a stochastic feature vector sampled at each time from a known distribution $\mathcal{D}_i$ (which can potentially varies over time). At each time, the decision maker observes both $x_i$ and a nosy observation of the rewards and decides whether to accept the reward (and then stop) or continuing the interaction. The objective is to attain a "good" competitive ratio w.r.t. the optimum a posteriori over the $n$ time steps. The authors propose two algorithm for the i.i.d. setting ($\mathcal{D}_i=\mathcal{D}$) attaining the $1-1/e$ competitive ratio, under a growth condition of the optimum. In the non i.i.d. setting, the author propose two algorithms. The first one attains $1/2$ competitive ratio against an optimum which considers the round from a fixed threshold to the end of the interaction, under a growth condition of the optimum. The second one assumes to have access to past samples (in the decision) and attains $1/2$ competitive ratio against the standard optimum.

**Strengths:**

Overall, I believe the results attained in this paper are interesting. Assuming that the rewards is linear in a latent vector is reasonable in practice and the authors effectively adapts techniques from contextual bandits to this "new" prophet inequality setting. The paper studies both the i.i.d. setting and the non i.i.d. one, thus extending the analysis to non-stationary settings.

**Weaknesses:**

My main concern is about the various assumptions which are made throughout the paper. Specifically,

1. It is not clear to me why the growth assumption should be either mild or reasonable in practice. To me, it seems just an artefact to overcome the theoretical hardness of the problem.
2. I believe this "new" contextual prophet inequality setting is interesting. Nonetheless, I have some concerns on the non i.i.id. setting. Indeed, the distributions can vary over time; still, they are known to the decision maker. Differently, the latent vector is unknown but fixed. Thus, in some sense, the non-stationarity of the problem is known.
3. Finally, I see the results on the adversarial setting as less meaningful than the i.i.d. ones. To me, the relaxed benchmark in Theorem 5.1. is really weak since it depends on the exploration phase of the algorithm. Moreover, the window access assumption seems again just a way to overcome the weakness of the previous result, since it cancels the inherent difficulty of the prophet inequality problem.

Minor: Line 361-362 I think the $\max$ should be inside the expectation.

**Questions:**

Could you elaborate on the assumptions?

Since I am not an expert on this literature I will be happy to increase my score if the answers turn out to be pretty positive.

---

> ### Author Response · Authors · 2025-11-20
>
> We thank the reviewer for the helpful comments and address each point below.
>
> - **Comment: It is not clear to me why the growth assumption should be either mild or reasonable in practice. To me, it seems just an artifact to overcome the theoretical hardness of the problem.**
>
> **Response:**  The growth assumption is *not* an artifact of our analysis; it is required by the **information-theoretic hardness** of prophet learning under noise. Under noise, no algorithm can guarantee any non-trivial performance in competitive ratio without some structural condition (Proposition 4.1).
>
> This assumption is unavoidable in noisy environments: if the **signal** (i.e., $OPT$) is smaller than the **noise level**, distinguishing good observations from noise becomes impossible, and no meaningful competitive ratio can be guaranteed.
>
> Unlike regret-based learning, where performance depends mainly on **reward gaps**, prophet inequalities compare directly to the absolute benchmark $OPT$.  When $OPT$ is vanishingly small compared to noise, we cannot achieve a meaningful multiplicative (ratio) guarantee.  Thus, the non-degeneracy assumption is a **structural requirement of competitive-ratio analysis**, not an artifact of our algorithm.
>
>
> More formally, regret guarantees typically satisfy, with noise std $\sigma$,
> $$
> OPT - ALG \le \sigma C.
> $$
> By dividing both sides by OPT,  it becomes
> $$
> 1 - \frac{ALG}{OPT}
>     = \frac{OPT - ALG}{OPT}
>     \le \frac{\sigma C}{OPT}.
> $$
> Thus,
> $$
> Competitive\\_ratio=\frac{ALG}{OPT} \ge 1 - \frac{\sigma C}{OPT}.
> $$
> When $OPT$ is extremely small, the factor $\sigma / OPT$ becomes arbitrarily large, making reliable estimation—and therefore any meaningful competitive ratio—impossible.
>
>
>
> The condition of $OPT = \omega(1/\sqrt{f(n)})$ (above from zero) for $f(n)=n^p$ or $\log^q n$ is typically satisfied in practice because  $OPT = \max_{i\in[n]} X_i$ naturally increases with $n$.  For instance, if $X_i \sim \mathrm{Unif}[0,1]$, then
> $$
> OPT = \frac{n}{n+1} \approx 1,
> $$
> which readily satisfies $OPT = \omega(1/\sqrt{f(n)})$ for $f(n)=n^p$ or $\log^q n$. Many natural distributions—such as all Beta distributions with fixed $\alpha,\beta>0$ (including uniform)—automatically satisfy this condition. The assumption fails only in highly degenerate cases.  Now we provide a more general characterization of when the condition may fail. For a general bounded  reward distribution with CDF $F$ on $[0,1]$, independence gives
> $$
> OPT = \mathbb{E}[\max_i X_i]= \int_0^1 \left(1 - F(x)^n\right)\, dx.
> $$
> If $OPT \to 0$, then equivalently,
> $$
> \int_0^1 F(x)^n dx \to 1,
> $$
> which implies that $F(x)^n \approx 1$ for almost all $x$; equivalently, the distribution of reward places nearly all of its mass extremely close to zero, creating an almost zero-signal regime.  Thus, the condition $OPT = \omega(1/\sqrt{f(n)})$ on  Corollary 4.3 excludes only such highly degenerate, vanishing-signal cases.

---

> ### Author Response · Authors · 2025-11-20
>
> - **Comment: I believe this "new" contextual prophet inequality setting is interesting. Nonetheless, I have some concerns on the non-i.i.d. setting. Indeed, the distributions can vary over time; still, they are known to the decision-maker. Differently, the latent vector is unknown but fixed. Thus, in some sense, the non-stationarity of the problem is known.**
>
> **Response:**  We first clarify the source of difficulty in the non-i.i.d. prophet setting.  Even when the stage-wise reward distributions are **known**, the non-i.i.d. prophet problem is already substantially harder than the i.i.d. case because the agent faces a **single irreversible stopping decision**: once a reward from a non-identical distribution is passed—even if it happens to be the maximum— it cannot be revisited. Classical results show that even with *full distributional knowledge*, the best achievable competitive ratio is only $1/2$.
>
> Our setting is strictly more challenging. In contrast to prior non-i.i.d. prophet work, the reward distributions in our contextual setting are **not** known; instead, they are determined by an **unknown latent parameter** $\theta$. Thus, the agent cannot compute the distribution-dependent thresholds without learning $\theta$. Although the non-stationarity pattern (i.e., the linear form) is known, its governing parameter is *unknown*, and learning it is essential for constructing any meaningful stopping rule. This makes our setting fundamentally different from the classical non-identical-distribution framework, where all distributions are known.
>
> Moreover, even in the **noise-free** case with linear structure and unknown $\theta$, Proposition 5.2 shows that the difficulty can be significantly worse than in the known-distribution case:
> $$
> \frac{\mathbb{E}[X\_\tau]}{\mathbb{E}[\max\_{i\in[n]} X\_i]}
> \le \min\left\\{\frac{1}{d}, \frac{1}{2}\right\\}.$$
> Thus, when the dimension $d$ is large, achieving the classical $1/2$ ratio against the full benchmark is information-theoretically impossible, even without noise.
>
> With noise, the challenge intensifies: early-stage rewards cannot be reliably estimated, and the agent cannot determine whether the maximum occurs during the initial rounds. Hence, without additional structural information, achieving a competitive ratio of $1/2$ against the full benchmark is unlikely in this contextual non-i.i.d. setting.
>
> Finally, we emphasize that the non-i.i.d. prophet setting is not directly comparable to non-stationary or adversarial online learning or bandit problems. The prophet problem centers on a single irreversible stopping decision and is evaluated via a competitive ratio, rather than cumulative regret minimization. This fundamental difference leads to substantially stronger informational requirements.
>
> Moreover, unlike in online learning, no non-trivial competitive ratio is attainable in the adversarial (nonstochastic) reward setting for prophet inequalities. In this sense, the non-i.i.d. stochastic setting considered in our work represents the challenging but meaningful regime, where non-trivial guarantees are still information-theoretically achievable.

---

> ### Author Response · Authors · 2025-11-20
>
> - **Comment: The relaxed benchmark in Theorem 5.1 is weak since it depends on the exploration phase of the algorithm. Moreover, the window access assumption seems again just a way to overcome the weakness of the previous result, since it cancels the inherent difficulty of the prophet inequality problem.**
>
>
>
> **Response:**  The weakness of the relaxed benchmark is not algorithmic—it reflects an **intrinsic information-theoretic limitation** of learning in non-i.i.d. prophet settings with *unknown* reward distributions. Proposition 5.2 shows that even in the **noise-free** case, when rewards follow an unknown linear structure, the achievable competitive ratio can be as small as $\min\\{\frac{1}{d}, \frac{1}{2}\\},$ which may lie strictly below $1/2$ when $d$ is large.
>
> Under noisy observations, the learner cannot accurately estimate early-stage rewards; thus, if the maximum occurs early, **no algorithm** can detect it reliably without dedicating an initial exploration phase. This information-theoretic barrier—not the benchmark design—explains why a benchmark that excludes the first part of the sequence is unavoidable. For instance, as shown in Proposition 5.2, for the noise-free case with unknown distribution under linear structure, by excluding first $d$ stages for the benchmark, we have $\mathbb{E}[X_\tau]/\mathbb{E}[\max_{i\in[d+1,n]}X_i]\le 1/2$.
>
> In fact, under the noise setting, the benchmark should naturally be defined as
> $$\mathbb{E}\left[\max_{i \in [g(n),n]} X_i\right],$$
> for some function $g(n)$. To obtain any non-trivial competitive ratio, one must ignore the first $g(n)$ stages where the maximum may occur but cannot be identified reliably under unknown distributions and noise. The dependence on $g(n)$ therefore **quantifies the portion of the sequence for which reliable learning is possible**, rather than reflecting a limitation of our algorithm.
>
> Moreover, if $g(n)$ grows sufficiently fast, then we may choose the exploration length $l_n$ such that  $l_n \le g(n)$ (and satisfying the theorem assumptions),
> ensuring that learning occurs entirely within the prefix already excluded by the benchmark. In this case, the ratio with benchmark $\mathbb{E}\left[\max_{i \in [g(n),n]} X_i\right]$
> **converges to the optimal value $1/2$**, matching the classical guarantee for non-i.i.d. prophet inequalities with *known* distributions. Thus, the relaxed benchmark is not weak; it is the **minimally necessary benchmark**, and our exploration schedule aligns with it to recover the optimal constant.
>
>
>
> Because of this inherent hardness, prior work has **never** addressed learning in prophet inequalities under unknown non-identical distributions without assuming access to **$\Theta(n)$** offline samples. For example, [Rubinstein et al., 2019] require $\Theta(n)$ offline observations to reach a $1/2$ ratio in the non-i.i.d. case; even for unknown i.i.d. distributions.  In contrast, in our linear-structured setting, only $l_n$ samples—potentially as small as $\log^p n$—are sufficient to recover the optimal $1/2$ competitive ratio under non-i.i.d. We expand on this point in Remark 5.5 and Appendix A.9.
>
> Regarding window access, the assumption does **not** remove the intrinsic difficulty. Proposition 5.3 shows that even with window size $w_n \le n-1$, the optimal competitive ratio remains **exactly $1/2$**; the problem does *not* become easier. Instead, the ability to revisit only the last $w_n$ rewards provides just enough minimal information to overcome the early-round uncertainty while preserving the fundamental hardness of the non-i.i.d. prophet problem.
>
> Moreover, window access naturally arises in many practical settings. For example, in hiring or admissions, evaluators may sequentially review candidates but are allowed to revisit only a limited subset of the most recently observed ones before making an irreversible decision.
>
> Overall, our results show that **minimal additional structure**—either relaxed benchmark or limited revisiting—is sufficient to overcome the impossibility barrier, and our analysis precisely characterizes how much information is required to obtain meaningful competitive ratios.
>
>
> - **Comment: Minor: Line 361–362 — I think the max should be inside the expectation.**
>
> **Response:**  Thank you for pointing this out. We've corrected this in the revised version.

---

### Official Review · Reviewer_KgWu · 2025-10-31

**Soundness:** 3
**Presentation:** 3
**Contribution:** 4
**Rating:** 6
**Confidence:** 3

**Summary:**

This paper studies the prophet inequality in a novel and practical setting where the underlying reward distributions are unknown, and the decision-maker only observes noisy realizations of the rewards. The authors propose algorithms based on Lower-Confidence-Bound (LCB) thresholding.

1.  For i.i.d. setting, the paper presents two algorithms, Explore-then-Decide (ETD-LCBT) and $\epsilon$-Greedy-LCBT, that both achieve the sharp $1-1/e$ asymptotic competitive ratio. This matches the optimal bound for the classical known-distribution case and is achieved under a mild non-degeneracy condition on the growth of the optimal prophet's value.
2.  For non-i.i.d. setting, the paper shows that the optimal $1/2$ competitive ratio can be achieved against a relaxed benchmark that excludes the initial exploration phase.
3.  By introducing a mild relaxation of window access, i.e., allowing the decision-maker to choose from a small set of past items, the proposed algorithm (ETD-LCBT-WA) achieves the optimal $1/2$ ratio against the standard, non-relaxed prophet benchmark.

Empirical results on synthetic datasets validate the theoretical findings, showing the algorithms achieve the claimed ratios and are robust to noise.

**Strengths:**

1.  The paper introduces a compelling new model for prophet inequalities that incorporates unknown distributions, noisy observations, and contextual features. This setting is well-motivated by real-world applications like hiring or ad allocation, where contextual data is available but true reward values are unknown and feedback is noisy.
2.  In this new, challenging learning setting, the paper achieves the sharp, classical competitive ratios of $1-1/e$ for i.i.d. and $1/2$ for non-i.i.d. with a window. It shows that the linear structure is sufficient to replace full distributional knowledge without sacrificing performance, given a mild non-degeneracy condition.
3.  The paper is very clear about why its assumptions are necessary. It explicitly provides impossibility results of Proposition 4.1 for noise and Proposition 5.2 for non-i.i.d. learning that motivate the need for its structural assumptions (the linear model) and conditions (the $OPT$ growth condition and the window-access relaxation).

**Weaknesses:**

1.  The sharp $1-1/e$ bound in the i.i.d. case (Corollary 4.3) hinges on the $OPT = \omega(1/\sqrt{f(n)})$ condition. The paper argues this is "mild" because $OPT$ is often bounded away from zero. While this is a reasonable assumption, it is the key condition that allows the algorithm to overcome the impossibility from noise shown in Prop 4.1. The paper could be strengthened by discussing this assumption in more detail, perhaps with examples of reward distributions where it might be violated, e.g., if $OPT \to 0$ quickly.
2.  To achieve the optimal $1/2$ ratio, the paper relies on either a relaxed benchmark (Theorem 5.1) or a relaxed problem setting (Theorem 5.4, window access). While both are well-justified, it leaves open the question of the standard non-i.i.d. problem against the full benchmark. The paper shows it's hard, but it's not clear if it's proven impossible to achieve $1/2$.
3.  The experiments are supportive but limited to synthetic data. While this is standard for a theoretical paper, the practical motivation would be bolstered by testing on semi-synthetic data, e.g., using features from a real-world dataset to generate the linear rewards.

**Questions:**

1.  The $1-1/e$ ratio in Corollary 4.3 depends on $OPT = \omega(1/\sqrt{f(n)})$. Could the authors elaborate on scenarios where this might not hold? For example, if the rewards $X_i \sim \text{Bernoulli}(p_n)$ where $p_n \to 0$ rapidly, thus $X_i$ are all extremely small, then $OPT$ might shrink too fast. How does this non-degeneracy condition compare to similar assumptions in the learning-to-price or contextual bandit literature?
2.  The paper shows that for the non-i.i.d. case, a $1/2$ ratio is achieved against a relaxed benchmark or the full benchmark with window access. Proposition 5.2 justifies the difficulty. Is it known or conjectured whether the $1/2$ ratio is impossible for the standard non-i.i.d. problem no window against the full $\mathbb{E}[\max_{i \in [n]} X_i]$ benchmark in this linear learning setting?
3.  The required exploration length $l_n$ depends on $L$ from Assumption 3.2. Remark 3.3 states $L$ may depend on $n$ and diverge. Remark 4.4 gives an example of $L=\sqrt{n}$. Could you elaborate on the practical implications of a diverging $L$? If $L=\sqrt{n}$, $l_n$ must be $\omega(\sqrt{n})$, which seems to fit the $l_n=o(n)$ requirement. Is this a common scenario, and does it present any practical challenges for the algorithm?

---

> ### Author Response · Authors · 2025-11-20
>
> We thank the reviewer for the helpful comments and address each point below.
>
>
> - **Comment: The $1 - 1/e$ guarantee in Corollary 4.3 requires $OPT = \omega(1/\sqrt{f(n)})$. In what scenarios might this fail? For instance, if $X_i \sim \mathrm{Bernoulli}(p_n)$ with $p_n \to 0$ rapidly, $OPT$ may shrink too quickly. How does this non-degeneracy condition relate to similar assumptions in learning-to-price or contextual bandit settings?**
>
>
> **Response:** First, without this additional assumption, no algorithm can achieve a non-trivial competitive ratio beyond zero under a noisy environment (Proposition 4.1). Thus, some lower bound on $OPT$ is information-theoretically necessary.
>
> Now we provide a more general characterization of when the condition may fail. For a general bounded  reward distribution with CDF $F$ on $[0,1]$, independence gives
> $$
> OPT = \mathbb{E}[\max_i X_i]= \int_0^1 \left(1 - F(x)^n\right)\, dx.
> $$
> If $OPT \to 0$, then equivalently,
> $$
> \int_0^1 F(x)^n dx \to 1,
> $$
> which implies that $F(x)^n \approx 1$ for almost all $x$; equivalently, the distribution of reward places nearly all of its mass extremely close to zero, creating an almost zero-signal regime.  Thus, the condition $OPT = \omega(1/\sqrt{f(n)})$ on  Corollary 4.3 excludes only such highly degenerate, vanishing-signal cases. In contrast, many natural distributions—such as all Beta distributions with fixed $\alpha,\beta>0$ (including the uniform distribution)—automatically satisfy this assumption.
> As you noted, if $X_i \sim \mathrm{Bernoulli}(p_n)$, then
> $$
> OPT = \mathbb{E}\left[\max_{i\in[n]} X_i\right] = 1 - (1 - p_n)^n.
> $$
> If $p_n \ge 1/n$, then $OPT$ is already a constant bounded away from zero, so our condition is immediately satisfied.
>
> This assumption is unavoidable in noisy environments: if the **signal** (i.e., $OPT$) is smaller than the **noise level**, distinguishing good observations from noise becomes impossible, and no meaningful competitive ratio can be guaranteed.
>
> Finally, unlike regret-based bandits or learning-to-price, where performance depends mainly on **reward gaps**, prophet inequalities compare directly to the absolute benchmark $OPT$.  When $OPT$ is vanishingly small compared to noise, we cannot achieve a meaningful multiplicative (ratio) guarantee.  Thus, the non-degeneracy assumption is a **structural requirement of competitive-ratio analysis**, not an artifact of our algorithm.
>
>
> More formally, regret guarantees typically satisfy, with noise std $\sigma$,
> $$
> OPT - ALG \le \sigma C.
> $$
> Dividing both sides by OPT, it becomes
> $$
> 1 - \frac{ALG}{OPT}
>     = \frac{OPT - ALG}{OPT}
>     \le \frac{\sigma C}{OPT}.
> $$
> Thus,
> $$
> Competitive\\_ratio=\frac{ALG}{OPT} \ge 1 - \frac{\sigma C}{OPT}.
> $$
> When $OPT$ is extremely small, the factor $\sigma / OPT$ becomes arbitrarily large, making reliable estimation—and therefore any meaningful competitive ratio—impossible.
>
>
> - **Comment: Is it known or conjectured whether the $1/2$ ratio is impossible for the standard non-i.i.d. problem (no window) against the full $\mathbb{E}[\max_{i \in [n]} X_i]$ benchmark in this linear learning setting?**
>
> **Response:**  **Proposition 5.2 establishes that achieving a $1/2$ competitive ratio against the full-information benchmark is fundamentally impossible in our linear model when the dimension $d$ is large, in the unknown non-i.i.d. setting *without* window access.**  Even in the **noise-free** case with linear structure, Proposition 5.2 shows that there exist instances where
> $$
> \frac{\mathbb{E}[X\_\tau]}{\mathbb{E}[\max\_{i\in[n]} X\_i]} \le \min\left\\{\frac{1}{d},\frac{1}{2}\right\\},
> $$
> which can be arbitrarily smaller than $1/2$ when $d$ is large.
>
> With noise, the difficulty only increases: reliable estimation of early-stage rewards becomes harder, and the agent cannot determine whether the maximum occurs during the initial exploration rounds.  Therefore, without relaxation, achieving a $1/2$ ratio against the full benchmark is unlikely.
>
> This motivates our results based on either (i) a relaxed benchmark or (ii) window access, both of which provide just enough structure to obtain meaningful guarantees.

---

> ### Author Response · Authors · 2025-11-20
>
> - **Comment: Remark 4.4 gives an example of $L = \sqrt{n}$.  Could you elaborate on the practical implications of a diverging $L$?
>   If $L = \sqrt{n}$, $l_n$ must be $\omega(\sqrt{n})$, which seems to fit the $l_n = o(n)$ requirement.  Is this a common scenario, and does it present any practical challenges for the algorithm?**
>
> **Response:**  The purpose of allowing $L$ to diverge is primarily to show that our framework is not restricted to the fixed-distribution setting considered in prior work, such as Goldenshluger \& Zeevi (2022). Instead, our assumptions accommodate scenarios where reward distributions may gradually evolve with the horizon, making the model more flexible from a *theoretical standpoint*.
>
> In practice, a diverging $L$ (e.g., $L = \sqrt{n}$) does not pose significant challenges as long as $l_n$ can be chosen so that $l_n = \omega(L)$ while still satisfying $l_n = o(n)$.  For example, if $L = \sqrt{n}$, choosing $l_n =C\sqrt{n}\log^p(n)$ meets both requirements.  Thus, the exploration phase remains a vanishing fraction of the horizon, and the algorithm continues to operate efficiently.
>
> The algorithm itself requires no modification; the only effect of a diverging $L$ is on selecting an appropriate $l_n$, which can be adjusted accordingly.
>
>
> - **Comment: The experiments are supportive but limited to synthetic data.**
>
> **Response:** We thank the reviewer for the thoughtful suggestion.  We agree that semi-synthetic experiments can provide additional practical insight.  At the same time, we note that empirical evaluation is **not standard** in the prophet-inequality literature—indeed, even synthetic experiments such as those in our paper are rare in prior work, including classical and modern results (e.g., Krengel & Sucheston, 1977; 1978; Samuel-Cahn, 1984; Chawla et al., 2010; Correa et al., 2017; 2019; Kaplan et al., 2020).  This is largely because prophet inequalities are inherently theoretical in nature, and empirical performance is typically not a primary focus.

---

### Official Review · Reviewer_g2Ym · 2025-10-31

**Soundness:** 3
**Presentation:** 4
**Contribution:** 3
**Rating:** 8
**Confidence:** 4

**Summary:**

This paper studies prophet inequality with linear-bandit-type reward: the $i$th true reward is $X_i = x_i^T \theta$, where $x_i$ is an observable feature vector distributed according to known distribution $D_{x, i}$, while $\theta$ is an unknown latent parameter in $R^d$.  The decision maker observes a noisy value $y_i = X_i + \eta_i$ each round, and needs to irrevocably decide to accept $X_i$ or continue.  The differences with classical prophet inequality are unknown reward distribution (because $\theta$ is unknown) and noisy observation.  The paper designs algorithms achieving approximation ratios $1 - 1/e$ for the iid setting and $1/2$ for the non-iid setting.  These bounds are claimed to be tight.  The main algorithmic idea is explore-then-decide: use ridge regression to estimate $\theta$ with samples from the first $l_n$ rounds; then use a classical thresholding algorithm for known-distribution prophet inequality but apply to the lower-confidence bound on the reward, computed from the estimated parameter $\hat \theta$.  Simulation results show that this algorithm achieves the desired approximation ratios and is better than heuristic algorithms.

**Strengths:**

(S1) This paper approaches the problem of prophet inequality with unknown reward distributions (which has been studied by some previous works) from a different angle: a linear reward structure with observable feature and unobservable latent parameter.  This is different from previous works (such as Correa et al, 2019) that assume unstructured unknown distributions.  This linear structure assumption allows the authors to prove very nice positive results: without any additional samples, by just going over the reward sequence once, the algorithm can achieve $1-1/e$ approximation ratio, which was achievable only with at least $\Omega(n)$ additional samples in the unstructured case.  Such a linear structure also allows noise in the reward observation.  This structural assumption is arguably realistic but has not been widely considered in the literature of learning-based optimal stopping or sequential search problems, so it is a good conceptual contribution to the literature in my opinion.

(S2) Besides the Explore-Then-Decide algorithm, the authors also analyze the performance of $\epsilon$-Greedy, the classical algorithm that mixes exploration and exploitation.  This makes the picture more complete.

(S3) The approximation ratios, $1-1/e$ for iid and $1/2$ for non-iid settings, are claimed to be tight.  (While these results seem to be true, I do have a question for the authors regarding formal proofs for these results.)

**Weaknesses:**

(W1) A small weakness is the lack of formal argument that the $1-1/e$ bound for the iid setting is tight.  The authors claim in Remark 4.4 that $1-1/e$ is tight, because Correa et al (2019) show that one cannot do better than $1 - 1/e$ if the number of samples from the unknown reward distributions is $O(n)$, _if the reward distribution is unstructured_.  However, in this paper, the reward distribution has a linear structure, and ridge regression can estimate the true reward parameter very accurately and efficiently.  So, I am not sure whether the previous $1-1/e$ negative result (for unstructured distributions) can imply the same negative result for the linear structured distributions. @authors: Can you comment on this?

**Questions:**

## Questions for the authors

(Q1)  See (W1).

(Q2) The iid experiment considers both ETD and $\epsilon$-Greedy, but the non-iid experiment only has ETD.  Why?  How does $\epsilon$-Greedy perform in non-iid setting?



## Suggestions

* Typo: line 125-126: $X_{\tau+1}$ should be $X_{n+1}$ ?

---

> ### Author Response · Authors · 2025-11-20
>
> We thank the reviewer for the helpful comments and address each point below.
>
>
>
> - **Comment: Tightness of $1-1/e$ in this setting.**
>
> **Response:** With full knowledge of the reward distribution and noise-free setting, the best single (quantile)-threshold policy is known to achieve a competitive ratio of $1 - 1/e$ (Correa et al., 2017).
>
> As noted in Remark 4.4, Correa et al. (2019) show that the optimal competitive ratio under **unknown** distributions is $1/e \approx 0.368$, while achieving the classical $1 - 1/e \approx 0.632$ tight ratio requires $\Omega(n)$ offline samples. This implies that without additional structural assumptions, no algorithm can surpass the $1/e$ barrier under unknown distributions.
>
> Surprisingly, under a linear structure and a mild non-degeneracy assumption on $OPT$—which is necessary since without it no algorithm can obtain a non-trivial guarantee (Proposition 4.1)—our method is able to approximate the optimal single-threshold via an LCB-based estimator and thereby achieve the tight $1 - 1/e$ competitive ratio, even under the unknown distribution and noisy observation.
>
> - **Comment: $\epsilon$-Greedy for non-iid setting**
>
> **Response:** In our framework, $\epsilon$-greedy forces exploration at uniformly random rounds. These randomly chosen rounds must be excluded from threshold-based decision-making because the agent is required to *continue* during exploration. As a result, the maximal reward may occur during one of these random exploration rounds, making it impossible for the algorithm to detect—and this loss cannot be recovered in the non-i.i.d. setting. This fundamentally limits any non-trivial guarantee in the **non-i.i.d. distribution setting** against a deterministic benchmark. Moreover, because exploration occurs at unpredictable positions, it becomes impossible to set a uniformly stable threshold for the decision rounds, as in Eq. (5), which is essential for analyzing expected reward in the non-iid setting.
>
> For this reason, we focus on the Explore-then-Decide (ETD) approach in the non-i.i.d. setting. Its pure exploration phase—where the agent always chooses *continue*—is confined **deterministically** to the early prefix, ensuring that no potentially maximal reward is lost due to random exploration. This structure enables non-trivial guarantees with a stable threshold under the deterministically relaxed benchmark or with a single window-access revisit. Guided by this theoretical insight, we therefore focus on ETD for the non-i.i.d. experiments.
>
> - **Comment: Typo: $X_{\tau+1}$ should be $X_{n+1}$.**
>
> **Response:** Thank you for pointing this out. We've corrected the typo in our revision.

---

> ### Comment · Reviewer_g2Ym · 2025-11-27
>
> Thanks for the response.  Let me clarify my question regarding the tightness of 1-1/e.  You claimed in the rebuttal that
>
> > With full knowledge of the reward distribution and noise-free setting, the best single (quantile)-threshold policy is known to achieve a competitive ratio of $1 - 1/e$ (Correa et al., 2017).
>
> Indeed, in the known-distribution iid prophet inequality problem, non-adaptive thresholding algorithms cannot do better than $1-1/e$.  However, Correa et al., (2017) (and Correa et al, 2019) also showed that adaptive algorithms can do better, achieving a better bound of $0.745$.
>
> Since your algorithm achieves $1-1/e$, I agreed that it is the best achievable result for non-adaptive thresholding algorithms.  But an adaptive algorithm might do better than that.  You claimed in the introduction that your bound is "sharp".  Please either clarify that this bound is sharp for non-adaptive algorithms, or prove that even adaptive algorithms cannot do better than $1-1/e$ in your structured unknown distribution setting.

---

> ### Author Response · Authors · 2025-11-28
>
> **Response:** We fully agree with your observation. In the classical i.i.d. setting with *known* distributions:
>
> - (*Non-adaptive*) Single-threshold algorithms are limited to a competitive ratio of $1 - 1/e$.  *Adaptive* algorithms can achieve approximately $0.745$ (Correa et al., 2019b, Correa et al., 2017).
>
> Before clarifying our sharpness claim, we distinguish two notions of a “single threshold’’:
> (1) a single **value** threshold, and
> (2) a single **quantile** threshold.
> In the i.i.d. setting, they coincide.
>
> We clarify that the ratio $1 - 1/e$ remains best *as long as the policy uses a single quantile-based threshold* in the known-distribution setting. Consequently, no policy within this class can achieve a better guarantee in *our structured unknown-distribution, noisy setting* either. This is precisely the class of policies our algorithms are designed to approximate with a fixed $1/n$-quantile.
>
> Our use of the term **“sharp’’** was intended specifically with respect to this class of **single quantile-threshold policies**.
>
> - If $\theta$ were known (oracle case), the optimal single-quantile threshold satisfies
>   $$
>   \Pr(x^\top \theta \ge \alpha) = 1/n.
>   $$
> - Due to the unknown $\theta$, our algorithms are plug-in estimators of this oracle rule while incorporating uncertainty through LCB adjustments. Although uncertainty makes the threshold approximate rather than exact, the **policy structure** remains that of a **single learned quantile threshold**.
>   In particular, our Explore-then-Decide method directly implements this non-adaptive single-threshold structure after the exploration phase.
>
> Finally, whether fully adaptive algorithms can exceed $1 - 1/e$ in our **unknown-distribution, noisy, linear-feature** setting remains an open question; we will state this explicitly in the manuscript.
>
> We thank the reviewer again for pointing out this subtlety, and we will revise the text accordingly to clarify the notion of “sharpness.’’

---

### Official Review · Reviewer_Y9sN · 2025-11-01

**Soundness:** 3
**Presentation:** 3
**Contribution:** 3
**Rating:** 4
**Confidence:** 3

**Summary:**

This paper studies a variant of the bandit problem, which the authors describe as a “prophet inequality.” The agent observes the context $x_i$ and its sampling distribution $D_{x,i}$ at each round $i$, but the only available actions are **go** or **stop**. Once the agent chooses **stop**, the received reward is evaluated based on its ratio against the outcomes of all $n$ rounds.

**Strengths:**

According to the authors’ claim, this paper is the first to study *bandit reward–based prophet inequality* and successfully achieves competitive ratios comparable to related results in the literature. The introduction of the *window access* setting seems conceptually reasonable and somewhat interesting.

**Weaknesses:**

### Weaknesses

1. From a *linear bandit* perspective, the problem seems rather simplistic. Hasn’t the estimation of $\hat{\theta}$ been extensively studied for a long time?
   1.1. The LCB bound appears very similar to what we typically observe in linear bandit results; in fact, it looks almost identical to the classic bound from OFUL.
   1.2. The setting assumes that the agent knows not only the context distribution (even future distributions $D_{x,i}$) but also that the reward probability $\alpha$ can be easily computed via the inverse CDF. There is no exploration strategy involved — the agent merely decides whether to **go** or **stop**. If the sampling is sufficiently uniform so that $\hat{\theta}$ can be well estimated (or if $\theta$ were known), the problem seems almost trivial.
   Particularly, the only dimension-related constant one can rely on under limited exploration is the minimum eigenvalue, and it seems that in Proposition 4.1, the result holds mainly because $\lambda$ is assumed to be independent of $n$ and $1/\lambda$ is sufficiently small compared to $n$.
   Taken together, it seems that the algorithm performs well simply because $n$ is large enough so that the dimension and minimum eigenvalue are no longer critical issues. Therefore, while the algorithm works as expected, it does not appear particularly *novel* or surprising.

Overall, while there is a long line of work suggesting that the topic has intrinsic research value, I personally do not find it very clear what aspect of the problem is *challenging* compared to standard linear bandit analysis.

### Clarity

1. What is $w$? It suddenly appears starting from Theorem 4.2 and seems to play a critical role, yet I could not find a clear definition. Also, the notation involving $l_n = o(n)$ and $l_n = w(\cdot)$ keeps appearing — could the notation be simplified?
   For example, in Theorem 4.2, it would be clearer to specify the minimal $n$ (e.g., $n > \dots$) for which $l_n \in o(n)$ holds.

**Questions:**

1. **Difficulty compared to linear bandits:**
   Since the action is only *stop*, the environment handles the exploration, and the estimation method seems nearly identical, in what sense is this problem more challenging than linear bandits?
   1.1. Why is the ratio necessarily $1 - 1/e$ in the i.i.d. setting?
   1.2. In the non-i.i.d. setting, why can’t the ratio exceed $1/2$ even with access to offline data? A mathematical explanation would be appreciated.

2. Regarding $\lambda$ (and $\lambda'$ in Section 5), what assumptions are made in relation to $n$?

3. Proposition 5.2 claims that a minimal amount of exploration is necessary. However, as $l_n$ grows, the upper bound of the ratio is expected to increase. Is $d$ the best possible scaling? Is there any lower bound result showing that no improvement beyond the authors’ $l_n$ choice is possible?
   3.1. It is also confusing that the authors effectively use $l_n$ offline data before starting the stopping process, yet in the offline setting the optimal ratio is $1/2$, while the proposed method achieves a higher value. How can this be reconciled?

4. Why must the rewards be non-negative? Is there any trick or condition in the proof that specifically requires “above-zero” rewards?

5. In the *Contributions* section, the authors claim that their setting is “the first scenario that only observes noisy rewards together with feature information and reward distributions are unknown.”
   5.1. However, according to line 123, the agent does observe $x_i$ in every round, and even $D_{x,i}$ is known. Doesn’t this statement overstate the novelty of the contribution?
   5.2. Then what exactly did previous works observe in addition to noisy rewards, and how does this work differ in that regard?

---

> ### Author Response · Authors · 2025-11-20
>
> We thank the reviewer for the helpful comments and address each point below.
>
> - **Comment: (Compared to linear bandits) Since the action is only *stop*, the environment handles the exploration, and the estimation method seems nearly identical. In what sense is this problem more challenging than linear bandits?**
>
>
>
> **Response:** The estimation step resembles linear bandits, but the decision problem is fundamentally different. Prophet inequalities require a **single irreversible stopping decision**, evaluated by a **competitive ratio**, not cumulative regret. This introduces challenges not present in linear bandits:
>
> *Exploration is not automatic.*  Although contexts arrive exogenously, the agent chooses between  **stop** (ending all learning) versus **continue** (sacrificing the current reward for information). Thus, exploration is *endogenous* to the stopping rule.
>
> *Stage-dependent value.*  The value of continuing decreases with time since  $\mathbb{E}[\max_{i \ge t} X_i]$ shrinks as the horizon shortens.  Linear bandits do not exhibit this horizon-dependent objective.
>
>  *Uncertainty directly influences the stopping threshold.*  With noisy and unknown distributions, the agent must determine whether its estimate is sufficiently reliable *before stopping*, unlike in classical prophet settings.
>
> *Our algorithms actively manage exploration for stopping.*  Exlore-then-Decide performs a **pure exploration phase**; $\epsilon$-Greedy distributes exploration **randomly across the horizon**.  Both rely on **LCB-based thresholding**, tailored to the stopping-time objective rather than regret minimization.
>
> We highlight that none of the prior literature on prophet inequalities addresses this online learning problem under both unknown and noisy reward distributions, despite such conditions being common in practice. Combining these elements—irreversible stopping, exploration under unknown distributions, noise, uncontrolled contexts, and a constant-factor prophet benchmark—requires analysis that goes beyond classical linear bandits and traditional prophet inequalities.
>
> - **Comment: Why is the ratio necessarily $1 - 1/e$ in the i.i.d. setting?**
>
> **Response:** In the i.i.d. prophet setting, $1 - 1/e$ is the **optimal competitive ratio** for any  single-threshold stopping rule under known distributions. The intuition is classical:
> - In an i.i.d. sequence of length $n$, each position is the maximum with probability $1/n$.
> - A threshold calibrated so that the stopping probability at each stage is $1/n$ yields
>   $$
>   \Pr(\text{stop before the end}) = 1 - (1 - 1/n)^n \to 1 - 1/e .
>   $$
> - This is the **best achievable competitive ratio** under a single threshold policy in the known-distribution, noise-free setting (Hill \& Kertz, 1982; Correa et al., 2017).
>
> Our contribution is to show that this **sharp ratio** remains attainable even when (i) distributions are unknown, (ii) observations are noisy (thanks to the structure considered). By constructing the stopping threshold from LCB estimates, both of our algorithms, ETD-LCBT and $\varepsilon$-Greedy-LCBT, achieve the same $1 - 1/e$ ratio under these significantly more challenging conditions.
>
>
>
> - **Comment: In the non-i.i.d. setting, why can’t the ratio exceed $1/2$ even with access to offline data? A mathematical explanation would be appreciated.**
>
> **Response:**  The $1/2$ barrier is **inherent to the non-i.i.d. prophet setting**, not a limitation of our algorithm. Even with full knowledge of the distributions, Samuel-Cahn (1984) shows that $1/2$ is the **optimal** competitive ratio for independent but non-identical rewards. Thus, offline samples—even unlimited ones—cannot surpass this bound.
>
> Interestingly, Proposition 5.3 shows that even with window access of size $w_n \le n-1$, no algorithm can surpass the $1/2$ competitive ratio. The reason is as follows:
>
> - The agent can never observe all $n$ rewards at the same decision point.
> - In the non-identical case, $X_1$ and $X_n$ may differ arbitrarily, making it impossible to infer one from the other.
>
> Thus, the $1/2$ limit reflects a **fundamental information barrier**: the gambler lacks the ability to compare all rewards simultaneously, whereas the prophet does. Our method matches this optimal bound.  Please refer to Appendix A.6 for the full proof of Proposition 5.3.

---

> ### Author Response · Authors · 2025-11-20
>
> - **Comment: Proposition 5.2 claims that a minimal amount of exploration is necessary. Is $d$ the best possible scaling?  It is also confusing that the authors effectively use $l_n$ offline data before starting the stopping process, yet in the offline setting the optimal ratio is $1/2$, while the proposed method achieves a higher value. How can this be reconciled?**
>
> **Response:**  First, our algorithm with window access does *not* achieve a ratio higher than $1/2$ in the non-identical distribution setting. The difficulty of this problem is proven in Proposition 5.2. We establish that even **without noise**, any algorithm cannot achieve a competitive ratio better than $\min\\{1/d,1/2\\}$ under unknown non-identical distributions. This already illustrates the intrinsic difficulty of the problem: when $d$ is large, obtaining a tight constant such as $1/2$ is impossible even in the noise-free case.
> However, if the prophet benchmark is relaxed by excluding the first $d$ samples for learning as $\mathbb{E}[\max_{i \in [d+1,n]} X_i]$, the achievable guarantee becomes  $\frac{\mathbb{E}[X_\tau]}{\mathbb{E}[\max_{i \in [d+1,n]} X_i]} \le \frac{1}{2}.$ Motivated by this (noise-free) result, in our **noisy** setting, we require an initial pure exploration phase of length $l_n$ to obtain reliable estimates (instead of $d$ in noise-free). Under the corresponding relaxed benchmark, our algorithm achieves  $\frac{\mathbb{E}[X_\tau]}{\mathbb{E}[\max_{i \in [l_n+1,n]} X_i]} \ge \frac{1}{2}.$
>
> To achieve $1/2$ for the standard prophet ($\mathbb{E}[\max_{i\in[n]}X_i]$), we consider window access. Here, window access simply allows the agent to revisit a limited number of recent past observations during the online process.
> The difficulty, even under the window access, is justified in Proposition 5.3. We show that even with window access of size $w_n \le n-1$, no algorithm can exceed the $1/2$ bound.  Our method achieves this optimal ratio (Theorem 5.4).
>
>  - **Comment: Novelty of the setting: unknown distributions with noisy rewards in prophet inequalities.**
>
> **Response:**  For unknown-distribution prophet problems, it is known that **no learning is possible** without additional offline samples (Correa et al., 2019a). Our contribution is to make learning feasible by introducing a **linear structural model**. This structure enables meaningful generalization that is unattainable in the classical prophet setting.
>
> We also emphasize that all existing works on (perfect-information) prophet inequalities consider the **noise-free** setting (Krengel \& Sucheston, 1977; 1978; Samuel-Cahn, 1984; Chawla et al., 2010; Abolhassani et al., 2017; Correa et al., 2017; 2019a; 2020).  Under noisy observations, however, **any non-trivial guarantee is not achievable** without any structural assumption (Proposition 4.1). Thus, noise fundamentally increases the difficulty of the problem.
>
> Our setting also differs fundamentally from standard online learning or bandit frameworks.  In bandits, the learner can repeatedly pull arms, progressively refine estimates, and ultimately optimize **regret**.  In contrast, prophet problems involve learning an optimal **stopping rule**, where each reward is observed only once and the decision to stop is **irreversible**, evaluated through a **competitive ratio**.  As a result, the value of “stop’’ versus “continue’’ depends jointly on the current observation, the remaining horizon, and the structure of the (unknown) distribution—creating a decision environment that is fundamentally different from bandits.
>
>
> - **Comment: Regarding $\lambda$ (and $\lambda'$ in Section 5), what assumptions are made in relation to $n$?**
>
> **Response:**  As stated in Theorem 4.2, the result holds under the pure exploration-length conditions  $l_n = o(n)$ and $l_n = \omega(L \log(d) / \lambda)$.  This implies that  $\frac{L \log(d)}{\lambda} = o(n),$  or equivalently,  $\lambda = \omega\left(\frac{L \log(d)}{n}\right) \text{as } n \to \infty. $
>
> In regret minimization, performance depends primarily on **gaps between actions** and remains stable even when the covariance matrix is poorly conditioned. In contrast, prophet inequalities require a **multiplicative guarantee** relative to the absolute benchmark $ OPT $. When the covariance matrix is ill-conditioned, the parameter estimation error becomes large, and this error directly affects the **threshold-based stopping rule**. Since the decision to stop is irreversible, even a small estimation error can cause the algorithm to miss the optimal reward entirely. Therefore, unlike regret settings, a **lower bound on $ \lambda $** is *information-theoretically necessary* to ensure that the threshold estimate is reliable and that the algorithm can achieve a **non-trivial competitive ratio**.

---

> ### Author Response · Authors · 2025-11-20
>
> - **Comment: Why must the rewards be non-negative? Is there any trick or condition in the proof that specifically requires “above-zero” rewards?**
>
> **Response:**  The non-negative reward assumption is standard throughout the prophet-inequality literature (e.g., Samuel-Cahn, 1984; Hill \& Kertz, 1982; Correa et al., 2019a, 2020; Immorlica et al., 2023,...).  Allowing negative rewards would make the competitive ratio meaningless as a performance metric:  if $\mathbb{E}[\max_i X_i] < 0$, then a smaller (more negative) $\mathbb{E}[X_\tau]$ would produce a *larger* competitive ratio, which does not correspond to better algorithmic performance.  To ensure that the competitive ratio remains a well-defined and meaningful measure, we follow the conventional non-negative reward setting.
>
> - **Comment: What is $w$ in Theorem 4.2? In Theorem 4.2, it would be clearer to specify the minimal $n$ (e.g., $n > \dots$) for which $l_n \in o(n)$ holds.**
>
> **Response:**  In Theorem 4.2, $w(\cdot)$ was intended to denote the little-omega notation. To avoid ambiguity, we will replace it with the standard symbol $\omega(\cdot)$ in the revised version. That is,  $f(n) = \omega(g(n))$  means  $g(n) = o(f(n)).$
>
> Regarding the minimal $n$, our analysis is asymptotic as $n \to \infty$. Algorithm 1 requires  $l_n = o(n)$ and $l_n = \omega( \frac{L \log d}{\lambda})$ to ensure the guarantee in Theorem 4.2.  Together, these imply  $n = \omega( \frac{L \log d}{\lambda} ).$ More concretely, for any constant \(c > 0\), this can be written as  $n > c \cdot \frac{L \log d}{\lambda}.$

---

### Author Response · Authors · 2025-12-02

Below is a concise summary of our contributions and how we addressed the major concerns.


# Final Remark on Contributions



- It is known that **no learning is possible** without additional offline samples of $\Theta(n)$ (Correa et al., 2019a), or without access to **multiple repeated sequences** used in regret-minimization settings (rather than competitive-ratio guarantees) as in Gatmiry et al. (2024) and Liu et al. (2025).



- Our work provides the **first (online) learning-based prophet-inequality algorithms for unknown distributions with noisy observations**, focusing on competitive-ratio (CR) guarantees by introducing a *linear structural model*.



- We develop LCB-based thresholding policies with exploration that achieve the tight *$1 - 1/e$* competitive ratio in the i.i.d. case and the optimal *$1/2$* ratio in the non-i.i.d. case under minimal structural assumptions.

---

# Summary of How Major Concerns Were Addressed

**1. Conceptual Novelty \& Distinction from Linear Bandits**

- The prophet problem centers on a **single irreversible stopping decision** evaluated via a *competitive ratio*, rather than cumulative regret as in bandits.


- In our unknown-distribution and noisy-observation setting, exploration is **endogenous to the stopping rule**: choosing "stop" terminates all future learning. This tradeoff, absent in bandits (and in prior prophet models), creates a qualitatively different decision structure.



---

 **2. Sharpness of the $1-1/e$ Guarantee (i.i.d. case)**

- The “sharp” bound is with respect to **single quantile-threshold policies**, the class our algorithms target to learn.

- For this class, **$1-1/e$ is the best**, even with known distributions (Correa et al. 2019b, 2017).



---

 **3. Hardness and Optimality in the Non-i.i.d. Setting**

- Even with full knowledge of distribution, the best competitive ratio (CR) is known to be **1/2**.

- Under unknown distributions with a linear structure, Proposition 5.2 establishes a stronger hardness result: $
  CR \le \min\\{1/d, 1/2\\} $ even in the *noise-free* setting, which may be much smaller than $1/2$ in large $d$. By relaxing the prophet benchmark to exclude the initial rounds—where reliable estimation is information-theoretically impossible—we show that the optimal ratio of $1/2$ becomes attainable.


- Therefore, in our noisy setting with unknown distributions, excluding the initial exploration samples is **information-theoretically necessary**, rather than an artifact of the benchmark design.


- Proposition 5.3 shows that window access still yields the same optimal **1/2** bound, and our algorithm achieves this optimal ratio under window access with targeting the standard prophet benchmark.



---

**4. Necessity of the Non-Degeneracy Condition $(OPT = \omega(1/\sqrt{f(n)}))$**


- Without this assumption, no algorithm can guarantee a non-trivial competitive ratio under noise (Proposition 4.1).


- This assumption is unavoidable in noisy environments (evident from the $\sigma/OPT$ term in Theorem 4.2, where $OPT = \mathbb{E}[\max\_{i \in [n]} X\_i]$): if the **signal** (i.e., $OPT$) is smaller than the **noise level**, then distinguishing true high rewards from noise becomes information-theoretically impossible, and no meaningful competitive ratio can be guaranteed.


- The condition $OPT = \omega(1/\sqrt{f(n)})$ (e.g., $f(n)=\sqrt{n}$) is satisfied for any constant-valued $OPT$. Moreover, since $OPT = \mathbb{E}[\max\_{i \in [n]} X\_i]$ naturally increases with $n$, this assumption is automatically satisfied by many standard bounded distributions.


---

---

### Meta-Review · Area_Chair_4iMq · 2026-01-06

**Summary:**

Some reviewers like the problem definition, algorithm design, and theoretical results of this work. Reviewers also have concerns on problem setup, claimed theoretical results, limited synthetic experiments, and assumptions. After reading the author rebuttal, I find all concerns have been addressed.

Given all four reviewers’ evaluations, the authors’ detailed rebuttal, and the overall novelty and significance of the work, I recommend Acceptance. However, this work can still be improved by including some real-world experiments, and further clarifications made in the rebuttal in the main paper of the next version.

**Reviewer Concerns:**

Reviewers have concerns on problem setup, claimed theoretical results, limited synthetic experiments, and assumptions. After reading the author rebuttal, I find all concerns have been addressed.

**Reviewer Scores:**

Only Reviewer g2Ym had the chance to reply to the author rebuttal and raised some additional questions, which are found to have been addressed by authors' further comments, therefore, I think Reviewer g2Ym would maintain a solid score at 8. Unfortunately, all other three reviewers didn't reply to the author rebuttal. After carefully reading their reviews and the author rebuttal, I find the rebuttal could address all concerns so a unanimous decision towards accept can be achieved.

---

### Decision · Program_Chairs · 2026-01-26

Accept (Poster)